# The Journey of 1000 Leagues towards the Decontamination of the Soil from Heavy Metals and the Impact on the Soil–Plant–Animal–Human Chain Begins with the First Step: Phytostabilization/Phytoextraction

Cristina Hegedus [1], Simona-Nicoleta Pașcalău [1,\*], Luisa Andronie [2,\*], Ancuța-Simona Rotaru [1], Alexandra-Antonia Cucu [1] and Daniel Severus Dezmirean [1]

[1] Faculty of Animal Science and Biotechnology, University of Agricultural Sciences and Veterinary Medicine Cluj-Napoca, Mănăștur Street, 3-5, 400372 Cluj-Napoca, Romania
[2] Faculty of Forestry and Cadastre, University of Agricultural Sciences and Veterinary Medicine Cluj-Napoca, Mănăștur Street, 3-5, 400372 Cluj-Napoca, Romania
\* Correspondence: simona.pascalau@usamvcluj.ro (S.-N.P.); luisa.andronie@usamvcluj.ro (L.A.); Tel.: +40-766-326-030 (S.-N.P.); +40-748-481-180 (L.A.)

**Abstract:** Nowadays, there are a multitude of sources of heavy metal pollution which have unwanted effects on this super organism, the soil, which is capable of self-regulation, but limited. Living a healthy life through the consumption of fruits and vegetables, mushrooms, edible products and by-products of animal origin, honey and bee products can sometimes turn out to be just a myth due to the contamination of the soil with heavy metals whose values, even if they are below accepted limits, are taken up by plants, reach the food chain and in the long term unbalance the homeostasis of the human organism. Plants, these miracles of nature, some with the natural ability to grow on polluted soils, others needing a little help by adding chelators or amendments, can participate in the soil detoxification of heavy metals through phytoextraction and phytostabilization. The success of soil decontamination must take into account the collaboration of earth sciences, pedology, pedochemistry, plant physiology, climatology, the characteristics of heavy metals and how they are absorbed in plants, and in addition how to avoid the contamination of other systems, water or air. The present work materialized after extensive bibliographic study in which the results obtained by the cited authors were compiled.

**Keywords:** heavy metals; health risks; phytostabilization/phytoextraction; plants; condition; advantage; limitation; precaution

## 1. Introduction

### 1.1. Major Causes of Soil Pollution

The spread and persistence of heavy metals are permanent, and becoming a global problem because the consequences are reflected at the level of soil, air and water, threatening the health of the entire ecosystem [1,2].

There are an impressive number of sites, over five million, contaminated with heavy metals/metalloids whose values exceed the regulated levels [3], and this is one of the reasons why soil is considered a threatened natural resource [4]. In addition, heavy metals, those invisible enemies of the soil, with a toxic character, can bioaccumulate in the food chain via a soil–plant–animal–human process [5].

In the ranking of heavy metals, arsenic occupies the first place, closely followed by lead (2) and cadmium (7). Cobalt (51), nickel (57) and zinc (75) are in the first half of the ranking from the group of compounds considered priority substances (275 compounds), which can affect the integrity of human health and animals and their products [6].

Included in the category of contaminants with metallic properties are transition metals; lanthanides; actinides, including cadmium (Cd), chromium (Cr), lead (Pb), nickel (Ni), zinc (Zn), selenium (Se), copper (Cu), cobalt (Co), manganese (Mn), molybdenum (Mo), iron (Fe); and from the metalloids, arsenic (As) and mercury (Hg) [7]. Heavy metals are characterized by relatively high density (>5 $cm^{-1}$), a high number of atoms and high atomic weight (63.5–200.6) [8].

Some of these heavy metals are considered non-essential metals for plants. Arsenic (As), cadmium (Cd), chromium (Cr), nickel (Ni), lead (Pb), vanadium (V), tungsten (W), mercury (Hg-metalloid) and uranium (U–radionuclide) manifest toxic effects even if the concentration in the soil is reduced [9,10].

Other heavy metals, such as aluminum (Al) (the third most abundant element in the earth's crust), iron (Fe) and magnesium (Mg), are part of the natural mineral constituents of the soil and have a presence in the earth's crust at a percentage of 7.4–8.1%, 4.7% and 2.1% [11,12].

Zinc (Zn), copper (Cu) and molybdenum (Mo) are considered micronutrients; their presence in the soil in small quantities is indicated, but in large amounts they have a toxic character [13].

*1.2. Natural Sources of Soil Pollution*

The causes of environmental pollution with heavy metals can be the result of natural processes or human activities. The natural causes are the consequence of the degradation of minerals containing heavy metals in the form of sulfides, such as As Fe, Pb, Zn, Au, Ni, Ag (silver) and Co; others, in the form of oxides (Mn, Al and, Cu, Fe, Co), can exist in both forms [14]. The physical characteristics of the soil (cation-exchange capacity, soil pH, soil aeration); the degree of soil erosion, volcanic activity, forest fires [15,16]; meteorological conditions, and especially the effects of hail; and the intensity, quantity and frequency of precipitation and the action of the wind also contribute to soil pollution [17].

*1.3. Anthropogenic Sources of Soil Pollution*

Heavy metals such as Co, Cr, Cu, Mn, Mo, Ni, Zn, Pb, Cd appeared as a result of the so-called "excess of civilization" [18]. Increasing unprecedented industry practices, improper waste disposal and storage have led to soil contamination [19]. In many parts of the world the storage and management of solid urban waste is still a challenge [20], especially in areas where heavy metals/metalloids such as As, Cd, Hg are considered to have ecological risk, ranging from considerable potential to high risk [21,22].

Lead (Pb), considered an important indicator for assessing environmental pollution due to persistence and a high retention time in the soil of 150–3000 years [23], comes from the cosmetic industry, pigment production, pewter pitchers, toys, mint, building materials, batteries and gasoline [24,25]. Both Pb and Cd are among the most common heavy metals polluting the soil and affecting the environment as a result of mining activities or car exhaust emissions [26]. Anthropogenic activities carried out in refineries, the metallurgical and steel industry, the non-ferrous industry, the use of PVC as a stabilizer in various products, the burning of fossil fuels and the incineration of waste represent sources of soil enrichment in cadmium [27].

The presence of cadmium in the "aerial basin" in the form of vapors or particles linked to chlorides, sulfates, and oxides can be transported over long distances with the possibility of being deposited on soil or at the surface water level [28,29]. The study carried out in areas of southern Iraq by Al-Hamzawi, A.A. and Al-Gharabi, M.G. [30] indicates that soil polluted with Pb and Cd is predominant in industrial areas (electrical power plants, main roads and traffic, car repair garages) and less in areas intended for agriculture or residential areas.

Another study demonstrated the fact that plants considered medicinal, such as *Rumex acetosa* (sorrel) and *Taraxacum officinale* (dandelion), that grow near the roadside, acumulate Pb and Cd but also Cu and Zn in the leaves, which could harm consumers' health [31].

The petrochemical industry, chlor-alkaline industry, mining, painting, mercury-based instrumentation [32,33]; dental amalgams [34]; coal-fired power plants [35]; and the use of metalloid as raw material for gold extraction are sources of soil contamination with mercury [36]. Fertilizers, steel mills, mint, nickel-plated jewelry, the production of auto parts and electrical parts, the combustion of fuels and detergents, and surgical instruments are sources of soil contamination with Ni [37].

Chromium (Cr) comes from the manufacture of stainless steel, tanneries, textile dyes, pigments, ceramic glazes, refractory bricks and fly ash [38,39].

Emissions of Cd, Hg and Pb are still a major problem as a result of the activities carried out in the manufacturing and extractive industry sectors [40]. Military activities and shooting sports contribute substantially to soil contamination with As, Cu, Mn, Mo, Ni, antimony (Sb) and Zn [41].

The primary sources of soil contamination in agriculture investigated by Rai, P.K. et al. [42] and Srivastava, V. et al. [43] are the result of the use of irrigation sources that may contain Zn, Cu, Ni, Pb, Cd, Cr, As and Hg. Other causes of soil contamination include atmospheric deposition and the application of treatments to combat diseases and pests in agriculture [44]. The extensive application of Hg as a fungicide and pesticide has resulted in soil contamination with this metalloid [45].

Dependence on the use of metallo-chemical fertilizers, pesticides (primarily Cu, Zn, Cd, Pb and As), herbicides and fertilizers (Cr, Cd, Cu, Zn, Ni, Mn, Pb and F), especially those based on phosphorus (P), lead to soil pollution [46–48].

It should be noted that the presence of cadmium from phosphates fertilizers on agricultural surfaces largely depends on the rock from which it is manufactured: cadmium-free Russian Kola igneous phosphate rock, or rock from Morocco that contains Cd [49].

As a negative consequence of using P fertilizer, Jayasumana, C. et al. [50] indicate that chronic kidney diseases of no known etiology have occurred in areas of Sri Lanka where the land has been treated with P fertilizer, the main source of As.

In 2014, Jigau, G. et al. [51] pointed out that, although organic fertilizers have beneficial effects on crops, they can come with an additional intake of heavy metals. The manure and droppings from pig and poultry farms can contaminate the soil with As, Cd, Cr, Cu, Pb, Hg, Ni, Se, Mo, Zn, Tl and Sb [52].

Supplementing animal feed with additives containing essential elements such as Cu, Zn, Fe, I, Mn, Mo and Se, either due to antimicrobial activity or to promote the growth rate, can represent other sources of soil contamination through the subsequent use of solid excrement as fertilizer [53,54].

Inorganic pollutants Pb, Co, Cd, Ni, As and Cr, as well as organic pollutants existing in the soil, as a result of their persistent and non-biodegradable nature, have negative consequences on the deterioration of the ecosystem in the soil [55] and the microbial activities of soil flora and fauna, and on the deterioration of the groundwater quality, and have the capacity to accumulate in plants, organs and animal tissues with negative repercussions on the health of humans, in whom various diseases and disorders, even in relatively small concentrations, can be caused [56–60].

## 2. The Influence of Heavy Metal on Plants and the Human Body

### 2.1. Phytotoxicity of Heavy Metals and Impact on Plants

The negative impact on plants depends on the metal, the form or compounds of the metal, and the ability of plants to regulate or store the metal [61]. The toxic response in plants varies between different heavy metals because the heavy metals possess different sites of action [62]. The toxic level of plant exposure to heavy metals leads to physiological, metabolic [63], structural, biochemical and molecular changes in their tissues and cells [64].

Some of the phytotoxic manifestations include the generation of reactive oxygen species and reactive nitrogen species [65], the replacement of enzyme cofactors, transcription factors, the inhibition of antioxidant enzymes [66], ion transport imbalance, DNA

damage, protein oxidation and peroxidation of lipids [67,68], as well as affecting nutrient metabolism and influencing the absorption of essential macro and micro minerals [69].

The most common visual evidence of the stress given by the presence of heavy metals consists of the decrease in the growth capacity and reproductive capacity in plants [70–72], the speed of seed germination decreasing [73,74], photosynthetic pigments being reduced [75], and, as a result, the capacity for chlorophyll assimilation decreasing and the number of leaves decreasing, with chlorosis, leaf necrosis occurring, and curly leaves [76–78].

### 2.2. The Negative Impact of Heavy Metals on Human Health

Some heavy metals, such as Zn, can be eliminated from the human body in a few days [79,80]; others, such as Cd, remain for 16–33 years in the human body [81].

As a general characterization, heavy metals cause damage to the DNA structure [24,82–84]. Carcinogenic effects are produced by their mutagenic capacity [85,86]. Through the action of free radicals produced (OH, $H_2$, $H_2O_2$), cellular destruction occurs when they act at the lipids level and the loss of the functionality of cell membranes occurs when the action acts on proteins [87]. Heavy metals can affect the normal functioning of the brain, lungs, kidneys, liver and other organs [88–90]. Long-term exposure can lead to the progression of physical, muscular and neurological degenerative processes, and the onset of diseases such as multiple sclerosis, Parkinson's disease, Alzheimer's, and muscular dystrophy [44,58]. The weakening of the immune system can induce immune diseases such as Hashimoto's, Graves', lupus, rheumatoid arthritis and Sjogren's [91]. Delays in intrauterine development, intrauterine fetal deformities [90] and partial blindness are effects of Al, Cd, Mn and Pb ingestion [92], which can also accumulate in bone and fat tissue [42]. In addition to other diseases and cardiovascular, gastrointestinal and allergy effects, heavy metals also affect the reproductive sphere [42,93].

### 2.3. Food Safety Possibly Threatened by the Consumption of Edible Plant Products and Mushrooms as Source of Vegetable Protein

Vegetable products, fruits and vegetables, in addition to components such as proteins, vitamins, fibers, antioxidants, iron and calcium, may contain varying amounts of heavy metals, which may threaten the integrity of human health [94]. Contaminant limits are regulated in industrialized countries, but not in developing countries where rapid industrial development and demographic explosion, together with the lack of pollution control, have caused an enormous increase in the contamination of agricultural soils with heavy metals [95].

The sources of soil pollution, along with the accumulation of heavy metals in plants, especially vegetables, and this reaching the food chain, have been studied in various areas of the world (Table 1).

**Table 1.** Main sources of heavy metals and accumulation in products of vegetable origin.

| Species | Country | The Source of Heavy Metals | Accumulated Heavy Metals | Author |
|---|---|---|---|---|
| 1. *Capsicum annuum* (green pepper) and *Lactuca sativa* (lettuce) | Northern Ethiopia | Irrigated soil | Cu and Zn | [96] |
| 2. *Beta vulgaris subsp. Vulgaris* (Swiss chard) | | | Fe, Mn, Cr, Cd, Ni and Co | |
| 3. *Lactuca sativa* (lettuce) and *Solanum lycopersicum* L. (tomato) | | | Cd | |
| 4. *Capsicum annuum* (green pepper), *Solanum lycopersicum* L.(tomato), *Allium cepa* (onion) | | | Pb | |
| 1. *Allium cepa* (onion, shoots and leaves) 2. *Solanum tuberosum* (potatoes) 3. *Daucus carota* (carrot) | Greece | Irrigated soil | 1. Cr (VI), Ni (II) 2. Ni can also pass to potatoes, depending on the irrigation concentration of the two heavy metals, through cross contamination 3. The results did not prove that Cr and Ni can cross-contaminate carrot bulbs. | [97] |
| 1. *Lactuca sativa* (lettuce) 2. *Cichorium endivia* L. (endive) 3. *Triticum* (wheat) and *Oryza sativa* (rice) | China | Phosphate fertilizer, leakage of factory sewage | 1, 2. Cd it has a concentration 4 times higher in leaves than in roots and 20–30 times higher than the concentration in the soil 3. Cd is accumulated in grains | [98,99] |
| 1. *Spinacia oleracea* (spinach) 2. *Brassica oleracea* (cabbage) 3. *Solanum melongena* (eggplant) 4. *Daucus carota* (carrot) | India | Irrigated soil | Cd ($1.30 \pm 0.31$ mg kg$^{-1}$), Pb ($4.23 \pm 0.32$ mg kg$^{-1}$), Cu ($1.42 \pm 0.25$ mg kg$^{-1}$), Zn ($3.4 \pm 0.28$ mg kg$^{-1}$), Cr ($1.16 \pm 0.11$ mg kg$^{-1}$) and Ni ($2.45 \pm 0.86$ mg kg$^{-1}$) | [100] |
| 1. *Mentha piperita* (mint) 2. *Spinacia oleracea* (spinach) 3. *Daucus carota* (carrot) | India | Irrigated soil (wastewater) | 1. Fe, Mn, Cu and Zn 2. Fe, Mn 3. Cu, Zn | [101] |
| *Solanum lycopersicum* (tomatoes) | Romania | Experimental field | Cu > Zn > Pb; | [102] |
| *Brassica oleracea* (cabbage), *Solanum lycopersicum* (tomatoes) | Ethiopia | Soil | As, Pb, Cd, Cr and Hg (even if the concentration is below the tolerable limit/day there is a risk of intoxication) | [103] |
| 1. *Spinacia oleracea* (spinach) 2. *Solanum melongena* (eggplant) 3. *Cucurbita pepo* L. (pumpkin) | Pakistan | Sewage water | 1. Mn, Cr and Fe 2. Cd, Ni, Zn 3. Cu | [104] |
| 1. *Spinacia oleracea* (spinach) 2. *Spinacia oleracea* (spinach) > *Brassica oleracea* var. *italica* (broccoli) > *Solanum lycopersicum* (tomatoes) 3. *Spinacia oleracea* (spinach) > *Beta vulgaris* and (beetroot) > *Petroselinum crispum* (parsnips) | Serbia | Soil (farm producers) | 1. Cd, Pb (Spinach appears to have the highest accumulation of heavy metals) 2. Ni 3. Cr | [105] |
| 1. *Allium porrum* (leek) 2. *Petroselinum crispum* (parsley) 3. *Allium cepa* (onion) | Turkey | Soil | 1. As, Cu 2. Ni, Mn 3. Zn, Cd, Pb | [106] |
| *Solanum Tuberosum* L. (potatoes) | Turkey | Soil cause by roadside industrial places irrigating pesticides | High: zinc, copper, nickel Less: cadmium, lead, chrome | [107] |

**Table 1.** *Cont.*

| Species | Country | The Source of Heavy Metals | Accumulated Heavy Metals | | | | | | | | | Author |
|---|---|---|---|---|---|---|---|---|---|---|---|---|
| 1. *Solanum lycopersicum* (tomatoes leaves) 2. *Cucurbita pepo* (zucchini) | Italy | Airborne pollutants | 1. Cd, Cr, Ni, Sn, Zn, 2. Ni, Sn, Zn, Ba | | | | | | | | | [108] |
| 1. *Lycopersicum* (tomatoes) > *Allium sativum* (garlic) > *Solanum melongena* (eggplant) 2. *Allium cep* (onions), *Allium. Sativum* (garlic), *Solanum lycopersicum* (tomatoes) *and Solanum melongena* (eggplant) | Pakistan | Waste water and household wastes and the use of heavy duty vehicles to convey sand from the river Acid–lead batteries as waste dumped in the river | 1. Cu 2. Pb, Co | | | | | | | | | [109] |
| *Lactuca sativa* (lettuce) > *Allium cepa* (onions) > *Daucus carota* (carrots) | India | Polluted and degraded environmental conditions | Pb | | | | | | | | | [110] |
| *Malus domestica* (apple fruits) | Greece | Local geology, plus fertilizers, pesticides, fungicides and insecticides | As (0.05–0.2); Cd (0.01–0.1); Hg (0.001–0.008) Ni (0.05–0.7);Pb (0.01–0.46); Zn (1.1–10.3) | | | | | | | | | [111] |
| *Lactuca sativa* (lettuce), *Amaranthus* (amaranth), *Vigna unguiculata* (cowpea), *Oryza sativa* (rice) | China | Soil near by coal-fired power plants, thermal power plants | Hg | | | | | | | | | [42] |
| 1. *Lactuca sativa* (lettuce) 2. *Phaseolus vulgaris* L. (bean) | Hungary | Irrigated water containing sodium arsenate (0.1, 0.25 and 0.5 mg L$^{-1}$) | 1. As: root > stem > leaf > bean fruit 2. root > leaves | | | | | | | | | [112] |

| Species | Country | | ppm | | | | | | | | | [42] |
|---|---|---|---|---|---|---|---|---|---|---|---|---|
| | | | Cr | Mn | Ni | Cu | Zn | As | Sr | Cd | Pb | |
| *Beta vulgaris* L. (Spinach leaves) | Bangladesh | soil | <0.05 | <0.06 | <0.65 | 5.59 ± 0.33 | 112.24 ± 0.47 | <0.01 | 23.75 ± 0.23 | <0.06 | 0.98 ± 0.00 | |
| *Lycopersicon esculentum* L. (tomato) | | | 0.51 ± 0.03 | <0.06 | <0.65 | 3.62 ± 0.29 | 31.1 ± 0.43 | 0.05 ± 0.0 | <0.14 | <0.06 | 0.12 ± 0.00 | |
| *Raphanus sativus* L. (radish—root) | | | <0.05 | 0.87 ± 0.13 | 0.87 ± 0.13 | 4.45 ± 0.34 | 25.78 ± 0.46 | 0.05 ± 0.00 | 7.23 ± 0.28 | 0.65 ± 0.05 | 0.51 ± 0.06 | |
| *Phaseolus lunatus* L. (bean—fruit) | | | <0.05 | 25.95 ± 2.56 | 0.87 ± 0.13 | 5.91 ± 0.22 | 68.34 ± 0.44 | 0.05 ± 0.00 | <0.14 | <0.06 | 0.65 ± 005 | |
| *Daucus carota var sativus* L. (carrot—root) | | | <0.05 | <0.06 | <0.65 | 5.35 ± 0.31 | 45.28 ± 0.45 | 0.04 ± 0.00 | <0.14 | <0.06 | 0.72 ± 0.03 | |
| *Brassica oleracea* L. (cauliflower—inflorescence) | | | <0.05 | <0.06 | 0.94 ± 0.29 | 4.59 ± 0.35 | 42.05 ± 0.43 | 0.05 ± 0.00 | <0.14 | 0.16 ± 0.04 | 0.23 ± 0.00 | |

**Table 1.** *Cont.*

| Species | Country | The Source of Heavy Metals | | Accumulated Heavy Metals | | | | | | | Author |
|---|---|---|---|---|---|---|---|---|---|---|---|
| | | | | Cd | Pb | Ni | Co | Zn | Cu | Mn | |
| *Coriandrum sativum* (coriander) | Pakistan | Soil | mg/kg dw | 0.23 | 2.12 | 0.77 | 0.47 | 36.65 | 5.92 | 21.65 | [113] |
| *Allium cepa* (onion) | | | | 0.13 | 0.66 | 0.54 | 0.32 | 23.94 | 6.25 | 20.15 | |
| *Lycopersicon esculentum* L. (tomato) | | | | 0.14 | 0.46 | 0.89 | 0.22 | 16.77 | 4.77 | 14.46 | |
| | | | | V | Cr | Ni | Cu | As | Cd | Pb | |
| *Eruca vesicaria* (rocca) | Quatar | Soil Irrigated farms | mg/kg | 17.09 | 6.41 | 1.70 | 13.074 | 14.72 | 0.9 | 6.36 | [114] |
| *Coriandrum sativum* (coriander) | | | | 15.91 | 6.03 | 1.38 | 15.30 | 16.86 | 0.43 | 5.00 | |
| *Petroselinum crispum* (parsley) | | | | 16.25 | 6.26 | 2.19 | 17.97 | 16.60 | 0.51 | 5.46 | |

Almost all plants accumulate small amounts of Pb, because it is normally found in the earth's crust in a percentage of 0.002%; however, among the plants studied by Zulfiqar, U. et al. [115], it seems that *Brassica oleracea* L. var. capitata (cabbage) accumulates high levels of Pb (1.7 mg kg$^{-1}$). Among the cereals *Triticum aestivum* (wheat), *Zea mays* (maize), *Avena sativa* (oat), *Hordeum vulgare* (barley) and *Secale cereale* (rye), *Zea mays* accumulates 0.88 mg kg$^{-1}$ followed by *Avena sativa* and *Secale cereale* (0.64 mg kg$^{-1}$). In addition to Pb, *Brassica oleracea* var. botrytis. (cauliflower) and *Brassica oleracea* L. var. capitata have a high affinity for Ni but a low one for Cd and Cu.

Fruit-type vegetables *Pisum sativum* (peas), *Glycine max* (soybean) and *Cyamopsis tetragonoloba* (cluster bean) can be grown on soils contaminated with Cd because it is not absorbed in the edible parts, but these crops are not suitable for soils contaminated with Ni and Pb. Root vegetables such as *Daucus carota* (carrots) and *Raphanus raphanistrum* (radish) accumulate small amounts of heavy metals, but leafy vegetables such as *Spinacia oleracea* (spinach), *Amaranthus* sp. (amaranthus) and *Sinapis* sp. (mustard) accumulate both essential and non-essential metals: Cd, Ni and Pb. *Allium cepa* L. (onion) and *Solanum tuberosum* (potatoes) accumulate high amounts of Cd and Ni, and low amounts of Zn and Cu [116].

*Glycine max* L. can accumulate large amounts of Pb [117], and *Solanum melongena* (eggplant) can accumulate Cd, Pb and Mn; *Lycopersicon esculentum* (tomato) can have large amounts of Fe, Pb, Mn and Zn [118].

*Solanum tuberosum* L. belongs to the category of candidate plants for the bioremediation of soils contaminated with Pb [119]. The positive correlation between heavy metals in the soil and their content in potato tubers varies, however, depending on the cultivars [120]. Codling, E.E et al. [121] recommend eating peeled potatoes, because Pb and As are mostly found in the peels.

Although a value below 300 ppm Pb is considered safe for consumption and Pb does not accumulate in the fruiting parts of tomatoes, strawberries and apples, still, those can be contaminated due to deposits on plants in a greater proportion than the absorption of lead by the plants [122]. From cereals and legumes, wheat, corn, oats, beans and lentils, rice has the highest capacity to accumulate heavy metals [123].

Asgari, K. and Cornelis, W.M. [124] found that, in *Triticum aestivum* (wheat) and *Zea mays* grains (maize), the concentration of Cd and Cr exceeds the safety limit and maize grains have the ability to accumulate large amounts of Cr. *Brassica oleracea* subsp. *capitata f. alba* has the ability to accumulate large amounts of zinc in cabbage heads, being considered a candidate for zinc phytoextraction [125].

*Solanum tuberosum* L., beans, fruits, cereals, especially rice, olive oil [126,127], *Brassica oleracea* and *Amaranthus oleracea* contain significant levels of mercury [128].

Among the fruit trees, peaches and apricots are very sensitive to the increase of arsenic in the soil and apple and pear are the least sensitive and cherry is intermediate, according to Torres, M. [129].

*Stevia rebaudiana* (candyleaf) accumulates large amounts of As, Cd, Cr, Cu, Fe, Mg, Pb, Se, Zn, Al, Ag, Co, Ca, Mn and Ni in flowers and edible parts such as leaves and stems [130]; *Taraxacum officinale* (dandelion) is recognized as a hyperaccumulator for Cd, Zn and Cu [131], and *Artemisia dracunculus* (tarragon) bioaccumulates Pb and Hg [132].

From the category of medicinal plants that can take up large amounts of heavy metals, Bağdat, R.B. et al. [133], Pirzadah, T.B. et al. [134] and Gawęda, M. [135] mention *Mentha arvensis* (mint—Cu, Zn), *Lavandula vera* (lavender—Pb, Zn, Cd), *Rosmarinus Officinalis* (Cd, Pb), *Matricaria chamomilla* (Cd, Zn, Pb), *Aloe vera* (Cr), marigold, hollyhock, carraway, garlic, *Rumex Acetosa* (garden sorrel: high affinity for Pb, Cd Zn, Cu, Mn, Fe, Cr, Ni) and *Cannabis sativa* (common hemp). The melliferous plants, *Sambucus nigra* L., *Hypericum perforatum* and *Tilia tomentosa*, accumulate Cd from the soil in the parts of the plants consumed as a tea infusion [136].

Boawn, L.C. and Rasmussen, P.E. [137] specify that the toxic potential of heavy metals in the above-ground parts of plants, including leaves and stems, varies according to plant species and occurs at values > 400 mg Zn/kg, Mn > 1000 mg/kg and Cu > 40 mg/kg.



A valuable source of vegetable protein for human consumption is edible wild species of mushrooms: *Armillaria mellea*, *Cantharellus cibarius*, *Coprinus comatus*, *Lycoperdon perlatum*, *Tricholoma portentosum*, *Suillus luteus* and *Xerocomus badius*, that can accumulate Hg, Pb, Cd and As. *L.perlatum* concentrates the highest amounts of Hg and As in the results obtained by Nowakowski, P. et al. [138]. Other edible mushrooms, such as *Boletus pulverulentus*, *Cantharellus cibarius*, *Lactarius quietus*, *Macrolepiota procera*, *Russula xerampelina* and *Suillus grevillei*, show a high capacity to accumulate Hg, Cd and in some cases Pb [139]. The presence of mercury in *Imlera badia*, *Boletus subtomentosus* and *Xerocomellus chrysenteron* is dependent on its amount in the soil, but among all species, *Boletus subtomentosus* can represent a real threat to consumer safety due to the high capacity to accumulate this metalloid [140]. The amount of Hg in *Pleurotus ostreatus* is also dependent on the degree of soil or substrate contamination [141].

### 2.4. The Potential Risk of Contamination of the Food Chain with Products of Animal Origin and Bee Honey

In small quantities, certain metals such as Mn, Zn, Fe or Mo are essential for the normal functioning of the human or animal body, but in large quantities they can become toxic and can accumulate in some organs and/or tissues together with other non-essentials [142]. Comparatively, among the organs with affinity for Pb accumulation, its values in the kidneys are slightly higher, in horses, cows, pigs and lambs, than in the liver, "the body's detoxification plant" [143]. Soil contamination with Pb affects the quality of feed administered to dairy animals, reaching, in this way, the food chain, through milk [144]. Arnich, N. et al. [145] found Ni, Cr, Pb, Hg and Ca in the content of cow's milk. According to Briffa, J. et al. [146], heavy metals may be present in products of animal origin such as Cr in meat, Co in meat, butter and cheese, Cu in liver, As in meat, poultry and dairy products, Cd in liver, Pb in red meat and Zn in lamb, beef and cheese. Research conducted by several authors demonstrates the potential for the accumulation of heavy metals in different organs, tissues, and edible products, both in farm animals and in wild ones.

The authors [147,148] found Pb and Cd in muscle tissue from cows, sows and even wild boars, Cu in the livers of wild animals and animal farms, and in beef, pork and broiler chicken both frozen and fresh meat [149–153]. A series of research [146,154–158] has highlighted the fact that, in the egg/yolk albumen, Cu, Zn, Ni, Cr, Pb, Cd, Hg and As are present in different amounts depending on the geographical area.

Not only products of vegetable or animal origin can contain heavy metals, but also honey and bee products. Most often, honey and bee products are considered detectors of air pollution. Pollen can be considered a bioindicator of environmental pollution, while honey is considered a detector of lead in time and space [159]. Depending on the area and harvesting period, the quality and composition of honey and bee products can be "enriched" with heavy metals and metalloids: Ag, As, Cd, Cr, Cu, Pb, Sn, Zn, Ni and Hg [160–163]. More recently, bees have come to the attention of researchers to highlight the links between soil–plant–bee body-beekeeping products [164]. Borg, D. and Attard, E. [165] demonstrate that there is an extremely positive correlation between soil contamination, the accumulation of stannum (Sn) and As in plants, and the amount of these heavy metals in honeybees and propolis. The research carried out by Bakhtegareeva, Z. et al. [166] highlighted the fact that, on an alkaline soil rich in Cu and As, all bee products, but especially bee bread, accumulate high amounts of these metals.

Tomczyk, M. et al. [167] believe that Cd migration is possible through the soil–plant–bee–honey food chain due to melliferous plants such as goldenrod and *Taraxacum officinale*, considered Cd accumulators. Bees, although they can avoid plants whose principles are toxic, are not able to detect heavy metals in plants except at high concentrations, when they were found to decrease bee visit duration at *Helianthus annus* (sunflowers). If this finding applies to all melliferous plant accumulators and hyperaccumulators, the negative impact would be reflected in the entire ecosystem by reducing pollination [168,169].

## 3. The Green Miracle, Plants and Phytoremediation

### 3.1. Phytoremediation and Soil Decontamination Mechanisms

Due to the sorption capacity of heavy metals, metallophyte algae [170], fungi, bacteria and plants have come to the attention of researchers [6,171–174]. The concept of phytoremediation ("phyto"—plant and "remediation"—restoration) was introduced in 1983 by Chaney [175].

The phytoremediation process aims at limiting–stabilizing, sequestering, assimilating, reducing, detoxifying, degrading, mobilizing and/or mineralizing contaminants using plants that, through certain parts, including root, shoots, tissues or leaves, naturally remove, transfer, stabilize, reduce or degrade contaminants from soil, sediment or groundwater [176].

Eco technologies that are friendly to the soil include phytodegradation, rhizodegradation and phytofiltration, which are specific mechanisms for the elimination of organic contaminants [177]. The phytoremediation solution known as phytodesalinization aims to remediate salt-rich soils through the ability of halophytes to accumulate large amounts of salt in their shoots, and at the same time allows the support and normal growth of plants [178].

Phytovolatilization (evaporation of certain metals through the aerial parts of plants) and rhizofiltration (filtration of metals from water through the root systems of plants) are also part of phytoremediation mechanisms [179]. Two phytoremediation strategies can be used to remediate soils contaminated with heavy metals and more: phytoextraction and phytostabilization [180]. Phytoextraction, also known as phytoaccumulation and phytoabsorption, is a mechanism that lends itself to heavily contaminated soil under acidic conditions, and is considered a cost-effective approach and an ecological method specific to inorganic contaminants (Cd, Pb, Zn), although DDT (Dichlorodiphenyltrichloroethane), PCB (polychlorinated biphenyls), radionuclides and organic compounds (oil) can also be reduced from soil and water [9,181–183]. In addition, phytoextraction, as in the case of the other mechanisms, helps not only to restore the quality of the soil, but also plays an important role in the aesthetic improvement of the area [184].

Phytostabilization (phytorestoration) addresses soil or sediment contaminated with heavy metals (Pb, Cd, Zn, As, Cu, Cr, Se), hydrophobic organics (PAH, PCB, dioxins, furans, pentachlorophenol, DDT, dieldrin), ammunition waste (2,4,6-trinitrotoluene or TNT and RDX), radionuclides (Cs137, Sr90 and U) and waste nutrients (ammonia, phosphate and nitrate) [185,186]. Phytostabilization is more of an isolation technique than a decontamination technique because plants stabilize contaminants in the soil, thus reducing their mobility and availability in the environment by converting metals from a more soluble oxidation state to an insoluble one and preventing the percolation of heavy metals into groundwater, the dispersion of pollutants in the environment or their introduction into the food chain; in addition, it prevents soil erosion [187,188]. Microorganisms such as bacteria and mycorrhiza from the rhizosphere improve the immobilization efficiency of heavy metals because they absorb part of them in cell walls, produce chelators and promote the precipitation process [189].

The authors Mahajan, P. and Kaushal, J. [190] believe that phytostabilization is effective in the case of fine-textured soils, and Salt, D.E et al. [187] specify that a sand/pearlite textured soil contaminated with moderate amounts of Pb can be decontaminated. In addition, there is the possibility of the stabilization of radionuclides if their level is low. Through the mechanism of phytostabilization and less phytoextraction, the soil characterized as being part of the silty-to-silty loamy texture, rich in organic content and with high alkalinity, can reduce from the above-ground parts of plants the elements Cd, Cr, Cu, Ni and Zn [191].

### 3.2. Plants and the Main Mechanisms of Contaminant Uptake

The plants chosen in phytoremediation are usually known for their versatility and ability to grow in harsh conditions where soils are contaminated [192]. In order to use the full potential of the plant in phytoremediation mechanisms, in-depth knowledge is

needed regarding the natural mechanisms of plants, including biophysical, biochemical and molecular mechanisms through which the absorption of metals is achieved, transport and exchange between cell membranes, distribution, sensitivity, hyperaccumulation and hypertolerance in different environments [193–195].

The "green liver" of the biosphere, plants, through their metabolic activities and the presence of antioxidant enzymes such as dehalogenase, oxygenase [196], superoxide dismutase, glutathione reductase, peroxidase or other non-enzymatic factors (ascorbic acid, tocopherol), can detoxify contaminated environments [197]. Kaushik, P. [198] is of the same opinion. The author specifies that, under certain stress conditions, plants are able to produce their own organic chelates such as polypeptides; phytochelation; polysaccharides; organic diacids; and malate, citric acid and gluconic acid, which can bind heavy metals.

Enzymatic activity is responsible for the accumulation or transfer of inorganic and organic pollutants into plant tissue, where they are transformed into less toxic metabolites and less bioavailable products [199]. Some plants have the ability to detoxify soil contaminated with several heavy metals simultaneously [58].

Plants possess two heavy metal detoxification systems. The first system is based on metallothioneins, common to all living organisms, which can reduce heavy metals in shoots by binding them to roots [200]. The second system is based on the synthesis of phytochelatins; substances specific to plants that bind Cd, Cu, Pb, and Zn, and at the same time allow the almost normal functioning of the cell.

The metals being isolated in vacuoles do not affect the plant's metabolic processes, which take place inside the cytoplasm [201]. For example, Thlaspi caerulescens, Ni hyperaccumulator, store the largest amount of Ni in vacuoles, which leads to increased tolerance to this heavy metal [202]. Apart from the affinity for the four heavy metals, Hartley-Whitaker, J. et al. [203] mention phytochelatins involved in the detoxification of soil contaminated with As.

*3.3. Pro-Phytoremediation Arguments*

Due to the potential toxicity and high persistence of metals, soils contaminated with heavy metals represent an environmental problem that requires an effective and affordable solution [204]. Although, with the exception of hyperaccumulators, most of the plants used in phytoremediation mechanisms have a metal bioconcentration factor lower than 1, which means that the reduction by up to 50% of the amount of heavy metals in contaminated soils lasts longer than human life [205], conventional methods of the depollution of contaminated soil (ex situ) through physical and chemical remediation (extraction, filtering of pollutants, vitrification, electrokinetic treatment, excavation and treatment of contaminated sites outside site, incineration, soil washing, soil vapor extraction and solidification, solidification and stabilization), which involve years of work and extraordinarily high costs, are limited from a technical point of view; they lend themselves to relatively small areas [206–208].

Phytoremediation represents the ecological approach to restoring the quality of soil and contaminated water, which exploits the biological mechanisms of the plant for the benefit of humans with the possibility of removing heavy metals, metalloids, such as Cd, Co, Fe), Hg, Se, Pb, vanadium (V), wolfram (W), Cr, Cu, Mn, Mo, Zn, radioactive isotopes (U238, Cs137) and Sr90 [209,210], synthetic organic compounds, xenobiotics, pesticides and hydrocarbons [211]. Through phytoremediation mechanisms, the natural structure and texture of the soil is maintained; the effect of soil erosion is reduced, the microbiology of the soil is improved and the energy used is renewable the plants use solar radiation in the chlorophyll assimilation process [212], biodiversity is maintained [213], a small volume of waste is produced and, more than that, the mechanisms are accepted by the general public [214].

Phytoremediation is a "less invasive technology" considered the "Green Revolution". It is easy to implement and maintain, friendly to the environment and restores the aesthetic appearance of the affected areas [213–217]. The plants used in this process not only

accumulate or transform toxic metals, but also help to preserve soil fertility by providing additional organic matter [218]. Heavy metals affect plant productivity, while healthy soil leads to high yields per hectare [219].

Phytoremediation is not an expensive technology, and it does not involve expensive engines or chemicals. It reduces the risk of contaminant dispersion in the surroundings through wind, rain, percolation and groundwater pollution [220], and is applicable for the decontamination of sites with several categories of pollutants [58,221]. It has long-term applicability, it can be used in the decontamination of areas where other remediation methods would not be cost-effective or practicable, and the time for decontamination of the soil is reduced [222]. The last statement is accompanied by some examples. *Pelargonium attar* has high potential for extracting Pb from soil, but under normal conditions decontamination of a calcareous soil containing 1830 mg Pb/kg would take 151 years, and 914 years to decontaminate a soil with a more acidic pH and 39,250 mg Pb/kg. According to Egendorf, S.P. et al. [223], the phytostabilization mechanism greatly reduces the soil decontamination period.

A study carried out on land contaminated with Pb originating from battery recycling activities, using as plants *Zea mays* and *Chrysopogon zizanoides* (vetiver) enriched with citric acid as a chelator, led to an increase in the content of Pb in the aerial parts of the maize of 14 times, and in vetiver 7.2–6.7 times, compared to control lots (172 mgkg$^{-1}$ Pb). In addition, only 7 days after the administration of citric acid, a decrease in the amount of soluble Pb was found which, according to the authors, Freitas, E.V.et al. [224], indicates the degradation of citric acid in the soil without remanence and danger of water contamination. Through this combination, the recovery period of soil quality would take 19–20 years (95–100 crops of maize and vetiver, respectively), compared to 116–104 years in the case of remediation without chelators.

Laboratory research undertaken by Li, J.T. et al. [225] aimed to evaluate plants with the capacity to accumulate Cd from the soil, capable of removing > 5%/year (designated as the threshold value) of the total metal in the contaminated soil using several plant species. The results demonstrated that *Thlaspi caerulescens* (hyperaccumulator) removed between 7.06 and 38.8% of Cd, but non-hyperaccumulator plants, *Oryza sativa* (rice: 7.17–15.3% Cd), *Zea mays* (7% Cd) and *Brassica napus* var. oleifera (5.5% Cd), can act as well or maybe better as hyperaccumulators due to their increased biomass which, in a 15-year interval, can extract 50% of the total Cd existing in the soil.

The conclusion of a study carried out on sludge contaminated with Ni, Cd and Zn using *Thalsi caerulescens*, radishes and rape was that 20–100 annual crops obtained under the most favorable conditions are needed to clean the soil contaminated by Ni or Cd, nine cultures of *Thlaspi caerulescens* are needed to remove Zn from 400 mg to 300 mg kg$^{-1}$, and a period of 13–14 years is needed for Ni and Cd removal [226]. Brown, S.L. et al. [227] mention that the same hyperaccumulator would have the capacity to totally reduce the amount of heavy metals in the soil in about 28 years.

*Averrhoa Carambola* (star fruit), used by Li, J.T. et al. [228] in Cd phytoextraction, is considered by the authors as a feasible option in cleaning contaminated soils, the plant being able to reduce the contaminant by up to 50% in 13 years. Planting *Populus* sp. (poplar) and *Salix* sp. (willows -phytoextraction) could restore the Lommel area (Belgium), polluted with moderate amounts of Cd (1 mgkg$^{-1}$) and Zn, within 12.5–25 years, according to Suman, J. et al. [58].

From an economics point of view, the costs of phytoremediation are lower compared to other classical remediation technologies (ex situ) [229,230]. For the phytoremediation of an acre of contaminated sandy loam soil to a depth of 50 cm, the remediation costs vary between USD 60,000 and 100,000 compared to using conventional excavation methods [231]. Additionally, for the same depth to decontamination, Gavrilă L. [232], estimates a cost of 30–50 USD/m$^2$, i.e., approximately 150,000–250,000 USD/ha compared to ex situ treatments, where the costs would amount to 0.99–4.2 million USD for the same surface.

In accordance with Doran, P.M. [233] and Sarma, H. [211], the by-product biomass resulting from phytoremediation can be further processed. Some metals extracted by the plants can be recovered from the ash, generating income from recycling or finding use for obtaining car parts [234]. Through more advanced technologies for extracting, metals such as Au, Ag, Pb, palladium (Pd), Ni, Co and radionuclides (U) can be recovered (phytoremediation) and reused (phytomining) [235–237]. The rehabilitation of areas contaminated with heavy metals by using bioenergy crops can provide benefits to local communities and farmers [238].

Plants such as *Jatropha* (purging nut) or *Brassica* sp. could be used for bioenergy production through different methods: incineration, gasification, anaerobic digestion or pure oil production [239]. *Zea mays*, an energy plant with the capacity to remove Zn from the soil surface (0.4–0.7 mg kg$^{-1}$/year), can be used to obtain 33,000–46,000 kW h of renewable energy and heat, substituting, in this way, non-renewable resources; in addition, $CO_2$ emissions are reduced by more than $21 \times 10^3$ kg ha$^{-1}$ y$^{-1}$, according to Meers, E. et al. [240]. *Eucalyptus* (blue gum) used in the phytoremediation of soil contaminated with As lends itself to obtain bioethanol, according to Fujii, T. et al. [241]. *Ricinus communis* (castor bean), with cadmium (Cd) absorption capacity, can be used in the production of biodiesel obtained from oil seeds [242]. In addition, according to Carrino, L. et al. [243], the product obtained would contain non-toxic amounts of Cd, Pb, Zn, Ni and Mn and the residues resulting from biodiesel could be used in biogas production and ethanol.

Among plants with high tolerance to Pb (>1000 mg kg$^{-1}$ soil), *Cyamopsis tetragonoloba* L. (guar) used in phytoextraction and *Sesamum indicum* L. (sesame) in the phytostabilization mechanism could be used to obtain biofuel [244]. Since the *Helianthus annuus* accumulates low amounts of heavy metals in the above-ground plant parts, the flowers and the seeds can be used to obtain energy or technical lubricants [245].

Although the main purpose of the cultivation of *Linum usitatissimum* L. (flax) consists of obtaining linseed oil, the plant as such can be used as a raw material to obtain fibers considered 100% safe, because heavy metals such as Cu, Cd, Pb and Zn accumulate in the order root > stem > leaves > seed > fiber [246]. *Canabis sativa* (hemp), among its multiple uses in obtaining wood fibers, cellulose or fodder, also has the ability to decontaminate soil contaminated with heavy metals (Cd, Pb and Ni), pesticides, crude oil and polyaromatic hydrocarbons. Biofuel obtained by distillation into ethanol can be used safely. Compared to flax, hemp grown on contaminated soils cannot be used as a raw material for clothing because most of the heavy metals accumulate in the leaves [247].

Ornamental plants, aromatic or medicinal plants are feasible in the phytoextraction mechanism because there is the potential for valorization of harvested leaves or flowers by obtaining perfumes or essential oils without the commercialized products needing to pay attention to the integrity of the human health. Through distillation methods, the products obtained do not contain heavy metals [9].

*Mentha piperita* (mint) and *Lavandula angustifolia* (lavender) grown in areas highly polluted with heavy metals near non-ferrous metal smelters (Cd, Pb, Cu, Mn, Zn and Fe) accumulated moderate amounts of heavy metals in the harvested biomass; these, however, were not found in the composition of the extracted aromatic oils in the results obtained by Zheljazkov V.D. et al. [248,249].

The accumulation of heavy metals in cruciferous plants, and especially *Brassica Juncea* (Indian mustard), stimulates the synthesis of the sulfur-based organic compound, glucosinolates. Depending on the pH of the soil, the presence of metal ions and additional protein factors, glucosinolates are hydrolyzed into secondary products such as isothiocyanates, thiocyanates, epithionitriles or nitriles which, used in the form of biofumigants, have a biocidal action. They combat parasites and phytopathogenic microorganisms, bacteria or fungi that attack crops [250].

Plants used in phytoremediation can provide essential trace elements such as Se, Fe and Zn to human and animal bodies [251]. A concept mentioned by Pandey, V.C. et al. [252] is that of biofortification, which increases the nutritional value of grains and

edible vegetables. Therefore, combining phytoremediation with biofortification would have the benefits of combating certain deficiencies and remediating the environment at the same time. Both concepts help medical research. By consuming plants capable of accumulating the microelements present in the soil, the homeostatic balance of the body is supplemented and restored [253,254].

In experimental conditions, *Zea Mays* was able to extract the microelements essential for plant development, Fe Zn and Mn, from soil experimentally contaminated with 2 g of FeSO$_4$, CdCO$_3$, and Zn, Mn, Pb and Cr, which indicates the plant as a potential candidate in the process of phytoremediation and biofortification [255]. *Daucus carota* (carrot) and *Brassica oleracea*, var. italica (broccoli) grown on soil enriched with organic Se obtained by harvesting, drying, and grinding *Stanleya pinnata* shoots used in the phytoremediation of areas contaminated with Se-laden agricultural drainage sediment after Bañuelos, G.S. et al. [256] can participate in Se biofortification through the consumption of these vegetables.

*3.4. Limitations of Phytoremediation*

Although using plants to remediate persistent contaminants such as heavy metals/metalloids may have advantages over other methods, there are limitations to the large-scale application of this technology [257]. The accumulation of heavy metals in plants depends on the metals present, their bioavailability in soils, soil properties, plant species, required nutrients, seasonality, geographic location and geological and environmental factors [181]. Cleaning the soil through various mechanisms of phytoremediation requires long periods because decontamination is strictly related to biological and plant growth cycles [258]. Therefore, to determine the phytotoxicity thresholds of heavy metals, laboratory studies must be performed on soil from contaminated sites [259].

In most cases, phytoremediation mechanisms can only remove contaminants located at depths similar to their roots, which is why they would not be able to solve the problem of contaminated groundwater. In addition, the toxicity and bioavailability of biodegradation products are not always known. They can accumulate in plant tissues and can then be released into the environment, mobilized in the groundwater, transferred into the air or reach into food chains [174,199,260,261].

Supplementation with amendments can raise costs or generate toxic waste products [262]. Soil porosity can be negatively influenced in the phytoremediation process by adding ammonium for nitrogen and phosphates for phosphorus supplementation, which can produce stable precipitates by reacting with minerals such as iron or calcium [263]. The more clayey the soil, the more the phytoremediation process is slowed down because it contains reduced amounts of oxygen and small amounts of biodegradable matter that negatively affect the presence of nutrients necessary for the activity of microorganisms [264].

Climate can limit plant growth and biomass production, thereby reducing the efficiency of the phytoremediation process [213]. There are endemic species for each area dependent on climate and soil pH. In Europe, for areas with metalliferous alkaline soils, *T. caerulescens* and *A. halleri* can be used, but they are not suitable for slightly acidic soils and humid climates, where *Avena strigosa* (bristle oat) and *C. juncea* have a higher potential for Cd accumulation than normal plants [265].

The decontamination of mine tailings requires different conditions depending on the climate: temperate or arid/semi-arid. In the phytostabilization mechanism the plants used in the semi-arid and arid zone for mine tailings, decontamination could suffer from the reduced efficiency of nitrogen and water use compared to the plants used in temperate environments where the phytostabilization efficiency is related to the availability of light and the existence of nitrogen from the soil. In addition, in the temperate zone, using the chelators in phytoextraction could contaminate groundwater [266]. Dry conditions induce drought stress and increase by two to three times the rate of Cd absorption in plants compared to plants that are not subject to this stress [99]. The maximum accumulations of Pb and Cu in the tissues of the above-ground plants were observed during the dry period in *Carduus nutans* L. (musk thistle) and *Taraxacum officinale*. Conversely, the reduction of

accumulation was observed in periods with short-term precipitation [267]. Air temperature is a major limiting factor in phytoremediation because, in some situations, the rate of uptake of metals by plants increases linearly with increasing temperature [268].

In other cases, *Festuca arundinacea* (tall fescue), an easy-to-grow, widespread, cold-season perennial plant often used for the degradation of PAHs and pyrene, as well as heavy metals, accumulates the maximum amounts of contaminants in the spring and autumn seasons when temperature values are lower [258]. Apart from the air temperature, the soil temperature at plant roots influences the phytoaccumulation of heavy metals—as emerges from the research carried out by Baghour, M. et al. [269]. Using *Solanum tuberosum* mulch and different temperatures in the root zone found that As, Ag, Cr and Sb accumulate in large quantities in the different organs of the plant at temperatures of 23 –30 °C.

The form and availability of heavy metals are dependent on soil moisture. The more stable form of Cd is found in soil with higher moisture content compared to moderately moist soil [270]. Even the degree of air pollution can affect the capacity to accumulate some heavy metals. Microparticles of Pb remain on the surface of the leaves under the form of precipitates, while Cu and Zn can partially penetrate the leaves [271].

The vegetation stage is another limiting factor. Research has shown that, in *Helianthus annus*, the concentration of accumulated Cd is higher in the early stage of growth than at the flower bud stage. Then, heavy metals are concentrated at roots level and in old leaves that play the role of a defense and tolerance mechanism, thus avoiding the accumulation of toxic levels in the apical tissues, the most active from a physiological point of view [272].

In the implementation of agricultural strategies for the decontamination of Cd-contaminated soil, Ji, P. et al. [273] believe that *Solanum nigrum*, also known as "night shadow" or "pig death", has a high potential for Cd extraction with the specification that the transfer factor and the lowest bioconcentration factor are found in the flowering stage. Conversely, the application of EDTA ($0.1$ g/kg$^{-1}$) and the maximum accumulation of Cd in shoots and roots were observed in the flowering phase of *Solanum nigrum* compared to the other vegetation stages in the research conducted by Sun, Y. et al. [274].

Another challenge in the success of phytoremediation is the competitiveness between plant species. In the decontamination of mining sites in France, rich in Zn, *Anthyllis vulneraria* (woundwort) accumulates large amounts of this heavy metal, but the conclusion was that the plant can be used only at the beginning of the soil decontamination process, when *Festuca arvernensis* (field fescue) or *Koeleria vallesiana* (somerset hair grass) are involved in the decontamination as well. Escarré, J. et al. [275] found that *A. vulneraria* is less competitive and disappears after flowering.

The genetic factor can limit the success of phytoremediation. Basic, N. et al. [276] found that wild populations of *T. caerulescens* collected from different areas of Switzerland showed a wide range of tolerance to Cd concentrations in the soil, but also the different capacity to accumulate heavy metals, which indicates the presence of different genotypes influenced by the variability of selective pressures (Cd concentrations in soil) and population characteristics.

Two of the hyperaccumulators of Cd and Pb, *B. Juncea* (80 genotypes tested) and *B. napus* (28 genotypes of rapeseed tested), have different affinities for the extraction of the two elements depending on the degree of soil contamination: moderate or polluted [277]. *Ricinus communis*, a bioenergetic plant with high biomass and high accumulation factor for Cd and DDT compared to other plants under co-contamination conditions, dependent on the genotypes studied (23) by Huang, H. et al. [278], showed differences between the accumulation or translocation of the two contaminants.

In the phytostabilization of a soil rich in Cd with the addition of pig manure, the results obtained by Thongchai, A. et al. [279] demonstrated that, among several *Tagetes erecta* (marigold) cultivars tested, only two cultivars, Babuda and Sunshine, showed 100% survival rate, bioconcentration factors > 1 for roots and high flower production.

The toxic effect on *Helianthus annus* and *Sinapis alba* L. (mustard), the decrease in biomass in the second year of research in soil enriched with Zn (400 mg Zn kg$^{-1}$ soil), led

to the conclusion that plants can be successfully used in soils moderately contaminated with Zn (up to 200 mg/kg), but not higher [280]. *Helianthus annus* demonstrated abilities to remediate soil contaminated with Pb and Cd in conditions where the values did not exceed 200 mg. Above these values, the fresh and dry weight of the plants decreased in the research carried out by Alaboudi, K.A. et al. [281].

*3.5. Relation: Soil—Heavy Metals*

The mechanism of phytoremediation involves several factors associated with the nature of the soil: alkalinity and hardness (Cu) [282], soil absorptive capacity, soil texture, cationic exchange capacity (where the higher it is, the sorption and immobilization of metals will be higher [269]), permeability, hydraulic conductivity, pore volume and pore size [283]. The selective absorption of low-density cations is influenced by redox potential. In alkaline soils, anionic metalloids Cr, As and Se predominate, the reduced absorption of cationic metalloids being influenced by the negative charge of the soil [284]. Increasing the redox potential under soil alkalinity conditions can transform Pb into more water-soluble forms, such as oxides and hydroxides [285].

The presence of heavy metals in the soil depends on the texture of the soil and its mineralogical composition. In a gold extraction area with clay loam texture, the authors Durante-Yánez, E.V. et al. [286] found Hg, Pb and Cd. Alluvial soils may have high concentrations of Pb, Cd and Zn [287]. Serpentine soil is characterized by a low calcium (Ca): magnesium (Mg) ratio, and nitrogen (N), phosphorus (P) and potassium (K) deficiency, but is rich in Ni, Co and Cr VI, which accumulate in the floral organs and leaves of plants [288,289]. Heavy-textured soils with a high proportion of clay increase the critical limit of Cd in the soil [290]. The phytoextraction efficiency of Ni is higher in clayey soil compared to sandy soil in combination with *Helianthus annus*, following research conducted by Lotfy, S.M. et al. [291].

According to Guerra Sierra, B.E. et al. [292], in more recent soils (alfisols and ultisols) the absorption of heavy metals is higher than that of older soils (oxisol), but soil mineralogy plays an important role in the absorption of Pb and Cd, regardless of the type of soil.

The results of the research undertaken by Quezada-Hinojosa, R. et al. [293] regarding the biogeochemical activity in the rhizosphere correlated with the bioavailability of Cd were associated with the soil type (Hypereutric Cambisols and Cambic Luvisols), the exchangeable fractions, carbonates, and the existing organic matter, and less with the plant species used (*Hypericum maculatum* (imperforate St John's-wort), *Alchemilla xanthochlora* (*lady's mantle*), *Cynosurus cristatus* (*crested dog's-tail*), *Ranunculus acris* (tall buttercup), *Dactylis glomerata* (*cocksfoot grasses*) and *Acer pseudoplatanus* (*sycamore*)).

A factor considered important in the success of phytoremediation consists of the content of heavy metals present in the soil, the speciation of the metal, and the metal itself [294]. The mobility and availability of metals depend on the form in which they are found, because only metals in a dissolved and exchangeable form in organic and inorganic components can be absorbed and used by plants [295].

Regarding the bioavailability of heavy metals/metalloids in soil, there are three categories: slight bioavailability (Cd, Ni, Zn, As, Se, Cu); moderately bioavailable (Co, Mn, Fe) and least bioavailable (Pb, Cr, U), which are not taken up and translocated into the harvested biomass [204]. In general, only a fraction of the heavy metals in the soil are bioavailable for uptake by plants. The strong binding of heavy metals to soil particles or precipitation renders a significant fraction of heavy metals insoluble, and therefore unavailable for uptake by plants [204]. However, even relatively bioavailable metals can be phytotoxic at levels greater than 200 mg/kg in soil [184].

Heavy metals in the soil, depending on their nature, are fixed to their organic or inorganic components [296]; therefore, an essential role in the phytoremediation mechanism is played by pH, the content in clay and organic matter. The last two factors determine the negative charge of the soil [98]. Using organic matter and pH modification influences the absorption capacity of heavy metals and controls the solubility and hydrolysis of metal

hydroxides, the solubility of organic matter, carbonates and phosphates, and the formation of ion pairs [117,126].

The addition of organic matter has as its main advantages the improvement of soil quality, low cost, and, through the humic acids produced, its ability to bind heavy metals or metalloids [297]. It retains, in particular, Pb and Cd in the superficial layer of soil (0–5 cm) due to the sorption capacity of the soil [298]. According to Figueroa, J.A. et al. [299], Cd and Zn ions have a relatively low affinity to organic matter, while Pb and Cu has a strong affinity to organic matter. Under these conditions, the toxicity of Cu increases with the reduction of organic matter [300]. Many metal cations are more soluble and available in solution at a low pH (below 5.5) [301]. Cd is highly mobile at a pH of 4.5–5.5 [302]; As is relatively mobile in soils with acidic pH, but is usually absorbed by clays, oxides, and hydroxides. The solubility of Ni decreases with the increase in pH, and in highly alkaline soils Hg occurs in a soluble form [41].

A condition not to be neglected consists of the presence of several heavy metals that can converge on the action of synergism or antagonism between them [191,303]. The availability of phosphorus (P) is stuck in conditions of soil alkalinity with high Na content, mine dust or lime waste, burnt lime, and hydrolysis products, and the presence of Se, As, Cr and Mo [304]. In addition, alkaline soils can block elements such as Zn, Cu and B [41]. Pb and Cd interfere with the absorption of nutrients by plants [69,89].

## 4. Phytoremediation Phytoextraction/Phytostabilisation

### 4.1. Common Requirements for Plants Used in Phytoextraction/Phytostabilization

The ideal plants used in soil decontamination should possess multiple characteristics, such as physiological adaptability to different climatic conditions and current agronomic techniques [305]; resistance to diseases and pests [306,307]; ease of harvesting and processing; and deep roots [17,308]. In the top rank of plants with the ability to decontaminate deeper soil layers, *Medicago sativa* (alfalfa: Zn, Cd, Ni Ba, Cs, Pb, Cu, Cr, PAH) holds the supremacy, with the root reaching up to 4.5 m, followed by *Linum usitatissimum* (Cd), up to 1 m; *Helianthus annuus* (considered pioneer: As, Cd, Zn, Ni, I, U, 226Ra, 238U, 90Sr, 137Cs) up to 50 cm; *Lolium perenne* (English ryegrass: 134Cs, 58Co −25 cm); and *Brassica juncea* (As, Cu, Cd, Cr (VI), 238U, Zn, Se, 137Cs, Ni -hyperaccumulator: 90–12 cm), according to Vangronsveld, J. et al., [309] and McCutcheon, S.C. et al. [310]. For the success of phytoremediation, the plants should have bioconcentration factor 20 and 10 tonnes/ha biomass, or bioconcentration factor 10 and 20 tonnes/ha biomass according to Peuke, A.D. et al. [205]. As a precaution, it is preferable to avoid using invasive plant species in order not to distort the ecosystem of the area [311] and affect the local biodiversity [183].

### 4.2. Plants and Phytoextraction

Hyperaccumulators are herbaceous or woody plants with an innate ability to absorb heavy metals 50–500 times higher than the average of plants without heavy metal accumulation capacity [269]. They possess tolerance to the toxic effects of the existing metals in the soil, which represents a limiting factor [218]. They accumulate and tolerate, without visible symptoms, high concentrations of heavy metals in shoots compared to non-accumulating plants [220]. Despite the toxicity of heavy metals, several plants grow in/or near contaminated soils and can exclude, accumulate or hyperaccumulate heavy metals and adapt to the conditions in that area [197,244]. Most of these are annual herbs, perennial shrubs, and trees which are part of the families *Brassicaceae, Fabaceae, Euphorbiaceae, Asterraceae, Lamiaceae, Scrophulariac*eae [189], *Proteaceae, Caryophylaceae, Tiliaceae, Rubiaceae* and *Myrtaceae* [58].

In 2007, Sinha, R.K. et al. [89] specified that, at that time, more than 300 species of plants and trees with the capacity to remediate contaminated soils were known. According to Sarma, H. [211], their number exceeded 500 species, starting with pteridophyte ferns and ending with angiosperms such as sunflower or poplar. In the work published by Reeves, R.D et al. [312], 721 species are mentioned. Most of the hyperaccumulating plants, about 523 species, actively participate in the decontamination of soil by Ni; 42 species in

the removal of Co; 53 species in the removal of Cu; 41 species in the decontamination of Se-rich soil; 20 species for Zn; 42 species for Mn; and 2 species for thallium (Tl). Cioica N et al. [313] enumerate a higher number of hyperaccumulators compared to 2017 for the decontamination of soil rich in Cd (10 species), and with 14 for Pb and approximately 20 hyperaccumulators for As. Among the plants that demonstrate the ability to hyperaccumulate As there are 12 species of ferns from the Pteridaceae family, according to Bergqvist, C. [314].

Mejáre and Bülow (2001), cited by MS Liphadzi et al. [315], divide plants into three groups depending on the metal for which they have an affinity for accumulation, namely Cu/Co, Zn/Cd/Pb and Ni; plants of Type I accumulate Al, Ag, As, Be, Cr, Cu, Mn, Hg, Mo, Pb, Pd, Type II Ni, and Type III radionuclides, hydrocarbons, and organic solvents after many authors [316]. A hyperaccumulating plant will concentrate more than >1 mg g$^{-1}$ (0.1%) As, Co, Cr, Cu, Ni, Pb, Sb or Se, or 1% (>10 mg g$^{-1}$) Zn and Mn of the dry weight of the shoots, 10 mg kg$^{-1}$ Hg and at least 0.01% Cd, i.e., >0.1 mg g$^{-1}$ [317].

Hyperaccumulators have bioconcentration, bioaccumulation factor ($C_{shoots}$ mg kg$^{-1}$/ $C_{soil}$ mg kg$^{-1}$) greater than 1 [250]. Those plants whose bioconcentration and translocation factor are >1 mg kg$^{-1}$ lend themselves to the phytoextraction mechanism [318,319]. As a condition, plants should cause repulsion when ingested by herbivores to avoid contamination of the food chain [15].

Some studies have shown that certain animals and insects will not consume the plants used in phytoextraction due to the unpleasant taste. Animal observations made in areas where hyperaccumulating plants *Alyssum bertolonii* and *Thalspi* were growing showed that cattle, sheep and goats avoided eating these plants [253]. Behmer S.T. et al. [320] believe that the plants used to detoxify polluted soils have created certain defense mechanisms against the attack of phytopathogenic insects, and according to Henry, J.R. [213] the seeds of hyperaccumulating plants are generally small and have no nutritional value.

The most popular species used in phytoextraction are *Brassica juncea* and *Helianthus annus* due to their fast growth rate, high biomass yield, and their ability to tolerate and accumulate metals and other substances [198]. Many researchers believe that the species belong to the *Brassica* family, *Brassica juncea* (Cd, Cr(VI), 137Cs, Cu, Ni, Pb, U, Zn), *Brassica napus* (turnip: Pb, Se, Zn) and *Brassica oleracea* (ornamental cabbage: 137Cs, Ni, As), are highly feasible in phytoextraction mechanism [321,322]. After the Chernobyl disaster in the 1980s, *Brassica juncea* and *Amaranth* cultivars were used for soil contaminated with radioactive Cs137 [323]. The ability of the *Brassica juncea* to transport Pb from the root to the shoots introduces the plant into the category of hyperaccumulators, having resistance to concentrations of 500 mg/L [253]. According to Ghosh, M. et al. [15], *Brassica juncea* is capable of removing 1550 Kg of Pb per acre, being used in induced phytoextraction. Ebbs, S.D. et al. [324] consider that *B. juncea* is more effective in removing Zn from the soil compared to one of the well-known hyperaccumulators, *Thlaspi caerulescens*. Although it concentrates only a third of Zn in tissues compared to *Thlaspi caerulescens*, *Brassica juncea* is considered to be a redoubtable plant in soil decontamination by obtaining a biomass 10 times higher than that obtained by *Thlaspi caerulescens*.

By comparing several species—*Brassica*, fern (*Pteris vittata*) and *Populus nigra* (poplar)— with Hg accumulation capacity from the soil, Li, J.T et al. [228], concluded that *Brassica juncea* can accumulate more than 1 mg Hg/g dry weight of the plant, compared to other species that accumulate only 0.2 mg Hg/g dry weight. The administration of rabbit manure biochar treated at temperatures of 450 and 600 °C, in combination with *Brassica napus*, represents a feasible solution in the decontamination of mining soils. The plant has the ability to accumulate large amounts of As and Se, but also Cr, Cu, Ni and Zn in its shoots. At the same time, it demonstrated the ability to stabilize As and Zn, but also total Se, Co, Cr, Cu, Ni, Zn and Pb at the root level in the results of research carried out by Gasco, G. et al. [325].

*Helianthus annus* is used for the phytoextraction of heavy metals: Pb, Zn, Cd, Ni, Cr, Cu, nitrogen (N), phosphorus (P), potassium (K), Mn, but also U. In addition, it "frees" the soil from the presence of polycyclic aromatic hydrocarbons (PAH) [257,260]. The results of the

research carried out by Ndubueze, E.U. [179] demonstrated the fact that the *Helianthus annus* can not only accumulate heavy metals, but also improves the degradation of pyrene and 2,4 DDT. Therefore, its detoxification potential resides in the simultaneous remediation of heavy metals, PAH and organochlorine insecticides from the contaminated soil. Boi, J. [253] mentions that the *Helianthus annus*, after one month, achieved the incredible performance of removing more than 95% of the U in 24 h, and can remove other radioactive elements such as Cs and Sr from groundwater.

The hydroponic growth of *Helianthus annus* has been used to absorb radioactive metals near the Chernobyl nuclear site in Ukraine, as well as near a U plant in Ohio [254]. In comparison to *Brassica napus subsp. napus* or *Chrysopogon zizanioides*, *Helianthus annus* accumulates large amounts of Cd and is more resistant to pests and diseases [326,327].

Apart from the two plants—*Brassica juncea*, considered the star of the *Brasicaceae* family, and the *Helianthus annus*, among the consecrated hyperaccumulators—authors such as Khan, A.H.A. et al. [328], Wang, Y. et al. [329] and Liu, Y. et al. [330] enumerate *Tagetes erecta* L. (aztec Marigold) ornamental plant hyperaccumulator of Zn and Cd, *Thlaspi arvense* (Cd), *Noccaea caerulescens*, formerly known as *Thlaspi caerulescens* (Zn/Cd/Ni), *Sedum alfredii* (hyperaccumulator Cd/Zn, Pb accumulator with high tolerance to Cu toxicity [331]), *Viola baoshanensi* (Cd), *Pteris vittata* (can accumulate more than 23 g kg$^{-1}$ As in shoots [332]), *Mirabilis jalapa (*four o'clock*)*, and *Impatiens balsamina* (garden balsam: Cr, Cd, Pb, Cu, Zn and As, accumulates more than 100 mg/kg Sn and TF > 1).

However, phytoextraction can also be achieved with the help of plants whose heavy metal absorption capacity is lower, but which are characterized by a fast growth rate with the production of large amounts of biomass, according to Guidi Nissim, W. et al. [333]. Schnoor, J.L. [188] mentions *Brassica napus subsp. napus*, *Hordeum vulgare* (barley), *Humulus lupulus* (hops), cruficers, serpentine plants (climbing pumpkin), *Urtica dioica* (nettle), *Taraxacum officinale* and *Fagopyrum esculentum* (common buckwheat). In the phytoextraction of Hg bioavailable from the soil, Gavrilă, L. [232] mentions *Hordeum vulgare*, *Triticum aestivum*, *Lupinus luteus* (yellow lupine) and *Cynodon dactylon (*Bermuda grass*).

For soil contaminated with Pb, it is necessary to find solutions to improve the bioavailability of Pb$^{2+}$ or to use plants with the ability to better translocate Pb$^{2+}$ in the portions that can be harvested. *Vicia faba* (faba bean) is a plant with great potential in the remediation of soil contaminated with Pb; it is an extremely tolerant species to this metal, and it grows normally even on soils with a high concentration of this heavy metal. Although it does not accumulate large amounts of Pb in the shoots, it compensates with a much higher biomass than a hyperaccumulator [334].

The native plants *Agrostemma githago* (weed), *Plantago rugelii* (blackseed *plantain)*, *Alliaria officinalis (*garlic), *Taraxacum officinale* and *Ambrosia artemisiifola (*ragweed) present on the surface of a soil contaminated with Pb (55,480–140,500 mg/kg) have demonstrated good extraction skills of it. *Taraxacum officinale* was able to extract 1059 mg/kg of Pb from the contaminated soil in the first crop and 921 mg/kg in the second crop. *Ambrosia artemisiifola* in the first crop extracted 965 mg/kg of Pb, and in the second crop more than 1232 mg/kg [213].

*Fagopyrum esculentum*, a plant with fast growth tolerant of poor environmental conditions, including low temperatures and low rainfall and barren soil, is the first Pb hyperaccumulator species in the *Polygonaceae* family with high biomass productivity. It can accumulate 8 g Pb kg$^{-1}$ dry mass in leaves and 3.3 Pb kg$^{-1}$ in roots, and is also considered an Al accumulating plant (Al: leaves) [335].

*Rumex crispus* (curly dock) is among the candidate plant species for the phytoextraction of Zn and Cd, with good efficiency, extracting 26.8 Zn and 0.16 kg ha$^{-1}$ Cd without the addition of chelator [336]. In the areas contaminated with Zn near the smelters, *Rumex acetosa* has the potential to accumulate more than 900 mg/kg$^{-1}$ Zn in both roots and shoots [337].

Barrutia, O. et al. [338] consider that the plant can be used in the phytoextraction and revegetation of metalliferous areas containing large amounts of Zn, for which it has an affinity, but also in soil decontamination of Pb and Cd (20,480, 4950 and 14 mg/kg).

After Adamczyk-Szabela, D. et al. [339], *Taraxacum officinale* can represent a valuable indicator of soil contamination being used to evaluate the bioavailability of heavy metals such as As, Br, Cd, Co, Cu, Cr, Hg, Mn, Pb, Sb, Se and Zn.

*Ricinus communis*, a metallotolerant plant capable of growing in heavily polluted soils, has fast growth, high biomass and adapts to many types of soil and climatic conditions [340]. It is considered a pioneer plant for degraded areas, requires little water and grows in wild, inhospitable places in many regions around the world, where other crops for biofuel production would not survive [341,342]. *Ricinus communis* is considered a plant tolerant to many heavy metals, including Cu, Fe, Mn and Zn; it is B-phytoextractant (phytoextraction of B) with the addition of peat as an amendment, and it is suitable for the remediation of soil contaminated with Cd and Pb [343].

Regarding the ability of *Ricinus communis* to extract Cd, it has been proven that the plant can extract larger amounts than *Brassica juncea* due to the significant amounts of biomass obtained both in and above the soil. *Ricinus communis* can grow well in areas contaminated with As. It accumulates in shoots up to 43.5 mg kg$^{-1}$ when exposed to 10-fold higher concentrations in solution, without showing toxicity symptoms such as dark brown leaves, necrosis in the tips and the edges of the leaves, or the death of the plant [344].

*Linum usitatissimum* L. is considered an accumulator for Pb, Zn and Cd and a Sr excluder [345]. The cultivars tested by Zhao, X. et al. [346], Y2I329 and Y2I328, demonstrate remarkable abilities in Pb accumulation (up to 5389 mg/kg Pb, respectively, higher than 1000 mg/kg Pb). Other results demonstrated the fact that flax can become a candidate for Cu soil decontamination, observing a high capacity to accumulate Cu simultaneously with the increase of its amount in the soil [246]. Crop rotation and possible intercropping can be an advantage in phytoextraction [347]. For example, *Linum usitatissimum* L. is one of the plants that are not self-compatible, which is why the 5-year rotation between crops must be respected [348]. Chitosan, a naturally biodegradable complexing agent, added at less than 1% and co-planting *Pteris vittata* and *Ricinus communis*, significantly increases the concentration of As in *Pteris vittata* leaves and decreases the concentrations of As and Pb in castor seeds compared to the monoculture of *Pteris vitata* in the research conducted by Yang, J. et al. [349]. Other research, and the results obtained by Liu, L. et al. [350], demonstrate the superior absorption abilities of Cd by co-planting *Nicotiana tabacum* (tobacco) and *Kummerowia striata* (Japanese clover) as a result of the decrease in soil pH.

### 4.2.1. Induced Phytoextraction and the Effect of Chelators in Soil Decontamination

Two types of phytoextraction are known: the continuous one (natural), in which plants "naturally" accumulate impressive amounts of heavy metals during their entire life (metallophiles), and induced phytoextraction, assisted by chelators, accelerators, which play a role in improving the solubility, mobility and accumulation of heavy metal ions through the formation of chelate systems. In this way, heavy metals become more easily assimilated in plant organs [15,351].

The use of chelating agent is warranted and necessary for alkaline soils [352], which is why research has focused on studying the phytoextraction mechanism in more detail by incorporating acidifiers to improve the success of this phytoremediation mechanism [276]. The synthetic agents used in induced phytoextraction are divided between non-biodegradable and biodegradable. The category of non-biodegradable chelators includes EDTA (ethylenediaminetetraacetate) and DTPA (diethylenetriaminepentaacetate), and the biodegradable ones are EDDS (ethylenediaminedisuccinate), NTA (nitrilotriacetic acid), MGDA (methylglycinediacetate), GLDA (N,N-dicarboxymethyl glutamic acid), and natural low molecular weight natural organic acids (NLMWOA): citric acid, tartaric acid and oxalic acid, which demonstrate high biodegradation capacity and allow the hyperaccumulation of metals: Zn, Cd, Cu, and Ni [225,353].

For Pb, the following chelators were tested: EDTA (ethylene-dinitrile-tetraacetic acid), CDTA (trans-1,2-cyclohexylene-dinitrile-tetraacetic acid), DTPA (diethylenetrinitrilo-pentaacetic acid), citric acid and malic acid. Research has shown that exposing plants to EDTA for a period of two weeks could improve the translocation of the metal into the plant tissue, as well as raise the overall performance of the phytoextraction [32]. Blaylock, M.J. et al. [354] demonstrated that the use of EDTA, DTPA and CDTA can participate in the accumulation of amounts greater than 10,000 mg/kg Pb in *Brassica juncea* shoots, but for a substantial accumulation of Pb (>5000 mg/kg) the concentration of synthetic chelators (EDTA, DTPA, CDTA) must exceed 1 mmol/kg.

In laboratory conditions, with contaminated soil taken from the area of a former gas factory, Tassi, E. et al. [355] reached the final conclusion that the plants used, *Brassica juncea* (As 0.7–19.3 mg/kg) and *Lupinus albus* (Pb: 2–625 mg/kg), demonstrated remarkable abilities in Pb accumulation under the action of EDTA and biammonium hydrogen phosphate (BAP) used in the phytoextraction of As, due to the competition between arsenate and phosphate at the absorption site.

The results obtained by Huang, J.W. et al. [305] demonstrate the ability of *Zea mays* and *Pisum sativum* to become true Pb accumulators of a soil contaminated with 2500 mg kg$^{-1}$ Pb by the addition of EDTA, which influences shoot Pb uptake from 500 mg/kg$^{-1}$ to more than 10,000 mg/kg$^{-1}$. The same authors mention similar results by using citric acid to improve U absorption. *Pisum sativum* could accumulate up to 95–88% of the added Pb (50 and 100 mg Pb kg$^{-1}$) in the above-ground parts under the action of EDTA in the research carried out by Hegedűsová, A. et al. [356]. Combining EDTA with citric acid can improve the absorption of Cd, Cr and Ni in *Helianthus annus* plants [209]. The chemical stability of EDDS for Pb is lower than other chelators, but it is biodegradable, has the effect of accumulating several heavy metals, and works well in the phytoextraction of Cu and Zn [357]. EDDS promotes the growth of *Ricinus communis* cultivars Zibo-3 and Zibo-9 cultivated on soils with 3.53 mg/kgCd and 274 mg/kgPb, unlike the application of EDTA or citric acid, which affects the biomass obtained [358].

MGDA (methylglycinediacetate) used in a quantity of 10 mmol/kg is more effective compared to EDTA or citrate in the extraction of Pb from contaminated soil in association with *Fagopyrum esculentum*. Tamura, H. et al. [335] calculated the extraction of 2% Pb/year at three harvests of the plant in the same time period.

Glutamic acid ameliorates toxicity induced by Cr and improves the morphological, physiological and biochemical characteristics of the *Helianthus annuus* plant, which is why Farid, M et al. [359] recommend this combination in phytoextraction. However, in their opinion, more in-depth studies related to the elucidation of the associated molecular and genetic mechanisms must be carried out.

Other chelators, such as nitrilotriacetic acid (NTA) or citric acid, actively contribute to the accumulation of heavy metals in plants [17]. Citric acid is of natural origin, biodegradable, and not toxic to plants. Moreover, its growth is not limited [360]. Citric acid has a low molecular weight and appears in the cellular vacuoles of the tissues of photosynthesizing plants, being excreted by their roots. In conditions of phosphorus (P) deficiency, *Lupinus albus* roots eliminate citric acid and studies have shown that this leads to a qualitative phytoextraction worthy of consideration [361].

For soils contaminated with U, organic acids such as citric acid, acetic acid and malic acid can be used as chelators. Citric acid has the best ability to mobilize U from the soil and increase its absorption in the tissues of *Brassica juncea* plants; therefore, Chang, P. et al. [362] suggest its use as a viable alternative to intensify phytoextraction. In experimental conditions, *Brassica napus* exposed in the absence/presence of citric acid demonstrates that this chelator can be used in the phytoextraction of Cu with positive effects on the capacity of plant photosynthesis, improving enzyme activity and antioxidant actions by reducing induced reactive oxygen species (ROS) [75].

On soils classified as humic and dystric cambisols, zinc in the form of Zn$^{2+}$ is mobilized under the influence of citric acid and, to a lesser extent, of tartaric acid, regardless of the dose

used. Following the results obtained, the authors Pérez-Esteban, J. et al. [363] concluded that citric acid is effective in extracting Cu from disused mining soils and old blende mines. A natural chelating agent with good results in the phytoextraction of Cu is the polyphenolic extract from grape seeds used in the study by Volf, I. et al. [260], which allowed the accumulation of large amounts of Cu in the aerial parts of *Brassica napus subsp. napus*.

In gold phytomining, with alkaline soil pH, amendments such as ammonium thiosulphate, ammonium thiocyanide, potassium bromide, and potassium iodide have good efficiency in the extraction of Ni and Tl [236]. The absorption and translocation of Hg in the aerial parts of the *Helianthus annuus* plant increase with the administration of cytokinin and ammonium thiosulfate [364]. For As to be translocated from the root of *Brassica juncea* plants to aboveground plant tissues, Pickering I.J. et al. [365] used, with good results, dimercaptosuccinate, a compound with similar chemical properties to phytochelatins, after which the translocation of As was 5-fold higher compared to the control group.

### 4.2.2. Limitations/Precautions in the Phytoextraction Mechanism

The sorption of metals in soil particles and their low solubility are limiting factors in phytoextraction [366], and the availability of heavy metals is reduced at a high pH [10]. The source of heavy metal pollution dictates the choice of plants because the uptake and transport of metals depends on the species and genotype of the plants used [367]. Using citric acid in the phytoextraction of Pb is not effective in combination with the *Fagopyrum esculentum* [368]. Although the addition of sulfur amendment increases the assimilation of Cd, Cr and Ni in plants, *Oryza* sp., *Zea* sp. and *Sorghum* sp. inhibit their assimilation [369].

Calcareous soil contaminated with Cu, Zn and Cd, and with the addition of nitrilotriacetic acid and sulfur amendment, even if it increases the solubility of metals in soil, is not a viable solution for soil decontamination in a short period, according to Kayser, A. et al. [370], because the accumulation factor in plants is only two to three times higher. Phytoextraction can be limited even if hyperaccumulators such as *T. caerulescens* or *Brassica* sp. are used when the soil is contaminated with several heavy metals. In this case, the phytoextraction of Zn and Cd can be affected by the toxic effect of Cu exerted on plants [371].

*B. juncea*, *Zea mays* can extract lead from contaminated soils through induced phytoextraction [372], but Blaylock M.J. et al. [373] specify that the effectiveness of phytoextraction using *Brassica juncea* is dependent on the vigor of the plants and the amount of dry matter produced per acre/year, which must be greater than 3 tons of dry matter during the years of conversion (3–5 years), during which time the plant can accumulate large amounts of Pb from the contaminated soil. In other words, if the soil has 500 mg/kg of Pb, the plant must have the capacity to accumulate 5000 mg/Kg to clean the soil in a few years.

Although, under laboratory conditions, assisted phytoextraction and the use of EDTA in split doses or single dose had good results, especially by single dose administration, in the field the absorption of Cd and Pb and biomass production were more reduced; hence, the final conclusion of the authors Neugschwandtner, R.W. et al. [374] was that *Zea mays* is not suitable for soil decontamination in a short period of time and, in addition, there is the possibility of groundwater pollution.

With all the advantages of applying EDTA (3 mmol/kg$^{-1}$ EDTA), which has strong chemical stability for Pb and is not expensive (USD 1.95 per kilogram) [375], it is a compound that microorganisms cannot degrade and, as a result, increases the risk of groundwater pollution [363,375] and can exert a toxic character on plants [188,376]. The problem of the contamination of groundwater with heavy metals is addressed, especially in acidic, sandy soils whose absorption capacity is low [377]. The application of EDTA before the germination of *Helianthus annus* seeds leads to sunflower seedling emergence and the reduction of dry weight as a result of growth depression [378].

Iodine is an effective agent for mobilizing Hg, and its bioavailability in the soil is low. Iodide volatilizes easily when it is oxidized to iodine, but as a negative consequence iodide can be toxic to plants if the concentration is too high [379]. Although the research of many

authors demonstrates good results in the use of natural low molecular weight organic acids (NLMWOA), the successive application of citric, malic and tartaric acid does not increase the efficiency of phytoextraction of Cu because, through the microbial degradation of the carboxylic acid that consumes $H^+$ and releases $OH^-$ and $CO_2$ ions, it increases the soil pH from the initial 5.5 to 7.7 only 96 h after application, and with it the availability of Cu [294].

A disadvantage of hyperaccumulating plants such as *Brassica juncea* (Cd, Zn), *Brassica oleracea (Pb, Zn-* spiked soil), *Berkeya coddii* (Ni, Co), *Allysum bertolonii* (Cu/Co, Zn/Cd/Pb and Ni), *Thlaspi caerulescens* (Zn, Cd) and *Thlaspi ergingense* (Ni) is the fact that they have slow growth and a small amount of biomass per hectare [280,363,380,381]. In phytoextraction, it is preferable not to use edible crops because heavy metals can enter the food and feed chain, threatening the integrity of human and animal health status. The biomass production of plants grown on contaminated soils decreases over time due to the depletion of soil nutrients or infections that may occur [382]. The incineration of plants could pollute the air and the soil, which is why the ash obtained will be stored in specially designed areas and treated as hazardous or radioactive waste depending on the type of contaminants present in the soil [9]. If it is not economically viable to extract the metals after combustion then the amount of resulting ash must be reduced, and *Verbascum hapsus* L. *(*mullein) hyperaccumulator for Cd with high biomass and high calorific value (19,735 kJ kg$^{-1}$) has this advantage [383].

Harvest time plays an important role in phytoextraction [347]. One of the hyperaccumulators, *Brassica Juncea*, must be harvested in the mature phase of growth to prevent drying, crushing, and the brittle effect that can become a secondary source of toxic substances, which reduces the possibility of obtaining a greater amount of biomass/ha [384]. To mitigate the ecotoxicological effects and potentially increased toxicity caused by the high concentration of soluble metals remaining in the soil after harvest, Bernal, M.P. et al. [385] recommend the administration of biochar or clay with an immobilizing action and counteracting adverse effects.

### 4.3. Plants and Phytostabilization

Unlike phytoextraction, in the phytostabilization mechanism plants must have a dense root system, in order to produce a large amount of biomass at the root level, and possess the ability to immobilize the contaminant and retain it in this part of the plant [386] through root adsorption or the precipitation/complexation/reduction of metals [387].

Plants must have a tolerance to high pH or salinity [211]. An advantage of phytostabilization is that the harvested biomass is not considered hazardous waste [388], because after the contaminants are sequestered in the root vacuoles it prevents the contaminant from leaking into the deeper layers of the soil, having a protective effect on groundwater contamination [389]. As a condition for successful soil decontamination through phytostabilization, plants can have a low translocation factor but a high bioconcentration factor [390]. The so-called excluder plants that do not fit in phytoextraction can become plants for soil detoxification in phytostabilization by limiting the translocation of metals in tissues with the accumulation of large amounts at the root level [391]. This category includes grasses that have a double effect: one is to remove heavy metals, and the second is to reduce erosion and stabilize the soil.

The plants that have been used in phytostabilization over the years are *Secale cereale*, *Festuca sp.*, *Festuca ovina* L., *Festuca rubra* (Zn, Cd, Pb, Cu) in a moderately acidic mine [392], *Festuca arundinacea* Schreb. (Cd, Pb, N, P, K, Zn, PAH, TPH, Cu), *Dactylis glomerata* (Cd, Pb, Zn), *Lolium perene* (perennial *ryegrass*: Cu, Ba, Cd, Pb, P, Al, PAH), *Sorghum halepense* L. (Johnsongrass: Al, As, Cs, Cu, Mn, Ni, U), *Triticum aestivum* L. (Ba, Cu, Pb, Zn, Cs) [393], *Agrostis capillaris* (common bent: Zn, Cd, Pb, Cu), *Agrostis stolonifera* (creeping bentgrass: Cd, Pb, Zn, As, Cu) [394], and *Agrostis castellana* (highland bent: As, Cd, Pb, Zn, Al hyperaccumulator) [395]. This category includes grasses that have a dual effect. The first consists of removing heavy metals, and the second in reducing erosion and stabilizing the soil.

*Lolium* sp., *Sorghum* sp. (cultivated worldwide: Africa, Asia, America, Europe, Oceania) [396] and *Festuca* sp. (a cool-season perennial found in Europe, Asia and North Africa, Japan, Australia, USA [397] which grows best in moist loamy environments rich in organic matter but also tolerates drought and is well adapted to a wide range of soils [398]) are wide-spreading grasses and adapted to different climatic conditions, warm or cold, and tolerant to exposure to trace elements [399]. *Agrostis tenuis* (colonial bent) and *Festuca rubra* (red meadow) grasses tolerant to heavy metals are used in commercial applications for the phytostabilization of soils contaminated with Pb, Zn or Cu [232]. Two varieties of *Agrostis tenuis* give good results in mine waste decontamination, cv. Goginan for acid Pb and Zn and cv. Parys for Cu mine wastes, while for calcareous Pb and Zn mine wastes, *Festuca rubra* cv. Merlin has good results [366]. *Festuca rubra* and *Poa pratensis* (Kentucky bluegrass) are two of the plants used experimentally in the phytostabilization of soil contaminated with Hg sampled from the area of a chemical plant, with good results in the research carried out by Sas-Nowosielska, A. et al. [400].

A field study undertaken near a metal mine rich in Cu, Pb and Zn allowed the conclusion that *Agrostis tenuis* can be used in the phytostabilization of acid mine waste, and *Festuca rubra* in that of calcareous mine waste [401]. The two authors specify the fact that, although the two types of grasses are tolerant to heavy metals and adapt, the basic condition for successful decontamination is determined by the pH value.

Along with grasses, leguminous plants including *Lupinus albus* (white lupin: Cd, As, Pb, [402]), *Vicia sativa* (common vetch accumulates several trace elements, e.g., As, Cd, Cu, Ni and Zn, and has good tolerance to Ni, [403]), *Trifolium pratense* L. (red clover: Zn), and *Trifolium repens* L. (white clover: Pb, B, Cu, As and PAH, widely distributed from the arctic to the tropics, but is best adapted to humid temperate climates [404,405]) can be successfully used in phytostabilization.

A promising plant in the decontamination of a calcareous soil, tolerant to Cd and Pb stress, is *Medicago sativa* [406]. In experimental research carried out in pots with soil taken from Pb Zn mines (pH-8.2), it was shown that the intercalation of *Lolium perenne* L. and *Medicago sativa* L. mitigates the inhibition of plant growth, increases the content of nitrogen and chlorophyll in shoots and roots, increases the enzymatic activity of saccharase and alkaline phosphatase and antioxidant activity, and reduces oxidative damage and lead absorption in forage plants [407].

Ornamental plants *Chysanthemum maximum* var. *Shasta* (*Shasta daisy:* Pb, Cd, Cu), *Calendula officinalis* L. (pot marigold: Pb, tolerant up to 400 mg/kg$^{-1}$ Cu in soil), *Iris germanica* L. (common flag: Pb), *Alcea rosea* L. (hollyhocks: Cd), *Euphorbia milli* (crown of thorns: Cr; [333]), *Tagetes erecta* (Cr and tolerant to high concentrations of Pb, Zn, and Cd; [408]), *Lavandula anqustifolia* (up to 40 mg/kg$^{-1}$ Ni in soil), *Silene vulgaris* (bladder campion: Ni; [409]), *Amaranthus tricolor* (Cd hyperaccumulators; [410]), *Calendula calypso*, (pot marigold), and *Cinnamomum camphora* (*camphor tree*: Cd) [411] can be successfully used in phytostabilization. Two other ornamental plants used in wastewater sludge decontamination, *Hibiscus Rosa-Sinensis* (hibiscus) and *Rosa* sp. (rose), have shown promising results during the 30 days of the experiment, accumulating at the root Fe > Mn > Zn > Cu, but the rose is preferred [412]. *Panicum virgatum* (switchgrass) and *Iris savannarum* (iris) can represent candidates in the phytostabilization of soil contaminated with As which, although have a reduced uptake ratio and translocation factor, compensate with the possibility of multiple harvests throughout the year [413].

*Nicotiana tabacum* lends itself both to phytostabilization and phytoextraction. It accumulates at the root level, Co, Ni and Cd, and in the leaves bioaccumulates large amounts of Cd and reduced amounts of Zn, Se and Hg [414]. These results are also confirmed by Angelova, V. [415]. According to Manoj, M.A.D.M.R and Ranjitha G.M.K. [416], *Nicotiana tabacum* is considered a promising plant for reducing heavy metals from e-waste and preventing environmental pollution.

Although *Rapistrum rugosum* (turnip weed) and *Sinapis arvensis* (wild mustard) have a low ability to translocate Pb from root to shoot, they demonstrate a high capacity to

absorb Pb from the soil through the root. The dry weight of plants does not decrease significantly at different levels of lead oxide treatments, 100, 200, 300, 400 and 500 mg Pb/kg soil, which is why Saghi, A. et al. [417] recommend the use of these species as suitable in phytoremediation technology.

For the phytostabilization of soils in semi-arid areas polluted with phosphate limestone wastes, rich in Cd, Cr and Cu, *Plantago afra* (sand plantain) is the perfect candidate in the results obtained by El Berkaoui, M. et al. [418]. *Artemisia artemisiifolia* L. extracts in descending order, Zn > Pb > Cu > Cd > As > Cr > Ni, in the research carried out by Čudić, V. et al. [389]. Inter-culture between local metallicolos plants (*Anthyllis vulneraria*, *Festuca arvernensis*, *Koeleria vallesiana*, and *Armeria arenaria)* significantly reduces Zn, Cd and Pb in leaves compared to plants grown in monoculture on a soil from a mining area, in the results of the research carried out by Frérot, H. et al. [419].

*Brassica juncea*, recognized as a hyperaccumulator, may be a tool worthy of consideration in the phytostabilization mechanism for Hg removal, for which it has been shown to have a high tolerance level (500–1000 mg/kg soil). Raj, D. et al. [420] concluded that *Brassica juncea* has a high potential for the phytostabilization of Hg without significant harmful effects on the plant when the amount of mercury does not exceed 1000 mg/kg Hg in the soil, the accumulation capacity in different parts of the plant being root > leaf > stem in the 2nd and 3rd months of the experiment and root > stem > leaf in the 1st month of the experiment.

### 4.3.1. The Amendments and the Their Role in Phytostabilization Mechanism
### Organic Amendments

Compost has a strong action for immobilizing metals, can accelerate the restoration of the vegetal carpet, especially in areas where there are chemical or physical constraints of the soil, improves microbial biomass, soil water holding capacity and cation exchange capacity and raises soil pH [421]. Since cow compost reduces the availability of heavy metals in the soil, the risk of crop contamination is reduced, which is a benefit because it does not affect the safety and security of the food consumed [422].

In the research carried out by Rizzi, L. et al. [423], it was demonstrated that *Lolium italicum* and *Festuca arundinacea*e are resistant to a high content of heavy metals and can be used as plants for the phytostabilization of abandoned mining areas in Italy which are rich in Pb and Zn, using compost as an amendment in a proportion of 10 and 30%. The amount of metals accumulated in shoots and roots decreases with an increasing amount of compost applied; however, plant development and total biomass were improved, which is a benefit. Compost obtained from green waste and assimilable waste from the catering sector increased the soil pH and had the capacity to reduce the Pb and Zn [424].

Urban solid waste subjected to the process of composting reduces the solubility of As, Cu, Pb and Zn; in ground alfalfa, composted leaves could lead to a decrease in the bioavailability of Pb, according to Gudichuttu, V. [425], and according to Hernandez-Soriano, M.C. and Jimenez-Lopez, J.C. [426], by using peat, hay and maize straw, heavy metals are mobilized according to the following scheme: Pb > Cu > Cd > Zn. Spent mushroom compost in combination with native shrub *Atriplex halimus* used in the phytostabilization of some mining areas had good results in mobilizing the metals Cd, Pb and Cu, the combination being recommended by Frutos, I. et al. [427].

Organic amendments such as poultry litter extract, sugarcane vinasse and humic acid gave good results in mobilizing Pb, Zn and Cu and reducing the concentration from the edible parts of *Amaranthus tricolor* [428]. Unlike other heavy metals, such as Cu, Cd, Pb and Cr, the application of compost on surfaces contaminated with As (cation) leads to an increase in leachable As in the soil, but using biosolid compost has good results in As adsorption [429].

Plowed soil with a loamy clay texture, originating from a military activity area, incorporating organic phosphorus (P) (class B biosolids) to a depth of 10–15 cm of the soil profile and planted with *Miscanthus (silvergrass)*, improved the soil microbial community and en-

zyme activity—acid and alkaline phosphatase and β-D-glucosidase, less arylsulfatase—and it decreased the accessibility and bioavailability of Pb [430].

Superior results through a better accumulation of heavy metals in the roots of *Lolium perene* were obtained by using organic amendments (cow dung—organic) compared to Calcinite + urea + PK14% + calcium carbonate (synthetic) on an area with soil moderately contaminated with Zn, Pb, Cd. In addition, increased biomass was obtained, soil properties were improved, and the activity and functional diversity of the soil microbial community was superior in the soil treated with the organic amendment [431]. Vermicompost is also an organic amendment used. The administration of vermicompost significantly reduces As III, the dominant being As (V), and decreased the concentration of Cu, Cd, Cr, Co, Zn and Ni in the soil in the research carried out by Huang, M. et al. [432].

Used as an improver or fertilizer, vermicompost demonstrates remarkable absorption and immobilization abilities of Pb along with Zn, Cu, Ni and Cd in the roots of *Helianthus annus* [433]; it stimulates shoot growth and biomass in *Sorghum* under moderate pollution with heavy metals in the order Zn > Cu > Cd > Ni > Pb. The immobilization of accessible forms of Pb, Zn, Cu and Cd in *Solanum Tuberosum* L. was highlighted in the research carried out by Angelova, V. et al. [434] by administering separate, organic amendments (compost 10%) and vermicompost (10%).

Although sewage sludge resulting from wastewater treatment and subjected to the composting process may contain Cu, Ni, Cd, Pb, Zn and Cr, its use as a fertilizer and in the cultivation of *Dactilys glomerata* demonstrated good biomass growth, soil pH increase and root accumulation of Ni, Pb and Zn simultaneously with the reduction of soil contaminants in the results obtained by Radziemska, M. et al. [435].

The addition of wastewater sewage sludge containing Cd and Hg in soil with a silt-loam texture and the cultivation of MxG (*Miscanthus × giganteus*) energy crop highlighted the fact that MxG can represent a candidate for phytostabilization and, in addition, that biomass does not decrease at moderate contaminations with the two heavy metals [207]. In the research carried out by Antonious, G.F. and Snyder J.C. [119], sewage sludge can represent a good fertilizer for the soil; it increases the pH value by 1.5 units regarding nutrients N, Ca and P, whose concentrations can reach a value similar to a super-phosphate fertilizer, and it traps pesticides such as trifluralin. *Brassica oleracea* var. italica and *Capsicum* (pepper) can be cultivated on this kind of soil, which accumulates Cu, Zn, Cd and Pb in the edible parts below the levels established by the U.S. EPA, Codex Standard 230–2001, Revision 1–2003.

Other Amendments Used in the Phytostabilization Mechanism

Conventionally, amendments such as lime, gypsum, phosphate fertilizers, sulfate carriers ($SO_4^{2-}$), organic matter and plant biomass such as *Miscanthus giganteus* have a role in maintaining soil health by improving its physical–chemical properties [436,437]. Both gypsum and lime raise the pH value and reduce the mobility and availability of Pb, Cu and Zn, with the specification that gypsum shows higher solubility than lime [438]. According to Kaninga, B. et al. [439], the decontamination of soil containing Cd, Cu, Ni, Pb and Zn by applying lime has better efficiency if the initial pH of the soil is lower compared to a soil with a neutral pH. The phosphate used in the extraction of Cr, Se and As has increased efficiency on acidic sandy soils [440]. In soil, Pb is bound to organic matter, iron oxides and clay, but sufficient amounts of phosphate necessary for plant growth react with Pb, forming a new compound unavailable for assimilation by plants [441]. Chloropyromorphite is an insoluble Pb compound resulting from the administration of phosphates amendments, phosphoric acid, calcium phosphates, and other fertilizers based on phosphorus (P), provided that the amount added is greater than the doses currently administered as fertilizers [442].

The re-vegetation of a disused gold mining area required amendments based on superphosphate to immobilize arsenic from the tailings, and among the cultivated plants *Hordeum vulgare*, *Lupinus angustifolius (*blue lupine) and *Secale cereale*, the combination phosphates/*Hordeum vulgare* had the best results; the As accumulated in the biomass was

126 mg/kg$^{-1}$ (shoots) and 469 mg/kg$^{-1}$ (roots) in the research carried out by Mains, D. et al. [443]. Although phosphate fertilizers are effective in reducing the availability of Pb, they are not as effective for As phytostabilisation, whose solubility increases if the soil contains both metals. The recovery solution consists of adding ferrihydrite or high Fe biosolids compost and high-surface-area iron (Fe) oxide to reduce them [444]. Using fertilizers based on phosphorus reduces the bioavailability of Cd; excess phosphorus fertilization reduces the absorption of Zn in plants, and its presence in sufficient quantities mitigates the harmful effect of boron (B) [444].

*Lolium perenne* L., *Festuca rubra* L. and *Poa pratensis* L. are the plants used to evaluate the soil–plant amendment interaction in the phytostabilization of soils contaminated with Pb, Mn and, respectively, Cu and Zn, using lime, phosphate and compost as amendments, individually and in combination. After calculating the bioconcentration factor, it was found that the application of lime reduces Pb and Mn in plants while phosphate decreases the amount of Pb in plants simultaneously with the increased Mn [445]. Lime reduces the exchangeable fraction of Zn, while phosphate application has an accelerating effect on exchangeable Cu [446].

The conclusion of the two studies was that the combined addition of amendments significantly decreased the mobile fraction of metals in soils on which grow *Poa* for Pb and *Lolium* for Mn, and in the phytostabilization of Cu and Zn on moderately contaminated acidic soils. The authors recommend the combined application of amendments and the use of *Festuca* and *Poa* as plants.

In the remediation of soils with high concentrations of Co and Cr, the application of sulfur and humic acid to reduce soil pH and increase the bioavailability of heavy metals in combination with *Linum usitatissimum*, is recommended by Shehata S.M. et al. [447]. The addition of granular sulfide to soil contaminated with Hg, especially from the areas near the Chlor-alkali plant, reduces the evaporation of the metalloid from the soil by forming an insoluble compound, HgS [448]. The administration of endogenous sulfur increases the accumulation of Cd in the root of *Fagopyrum esculentum* plants, and the addition of exogenous iron reduces the accumulation of Cd and increases the tolerance to Cd stress of *Solanum Nigrum* plants [302,449]. Because of the better absorption of Se *Triticum aestivum*) plants, the replacement of sulfate from N-P-K with a chloride base form has the advantage of avoiding competition between sulfate and selenate [187].

A benefit brought by the administration of iron oxides on soil with a sandy texture and almost neutral pH consists of reducing the bioavailability of As by immobilizing it in the soil, along with reducing the danger of accumulation in plants and then in the food chain [450]. At the same time, hydrous Fe oxide materials can absorb Pb and reduce its availability [444].

Phytostabilization Assisted by Mineral Sorbents

The addition of zeolite, chalcedonite besides limestone and dolomite can raise soil pH and extract some heavy metals through the phytostabilization mechanism [451]. Fly ash, spent mushroom substrate, silkworm excrement and limestone immobilize Cd; composted sewage sludge and cultivation of *Helianthus annuus* immobilize Cd and Ni; and compost and limestone dolomites limit the translocation of Cu, Cr, Zn and As [452]. It was experimentally demonstrated that using zeolite or limestone to reduce Cu had positive effects, these amendments having the ability to form stable complexes in the soil [451]. The results of the research carried out by Radziemska, M. [451] demonstrated that zeolite and chalcedonite can accumulate Cu in the root, simultaneously with the reduction of toxicity in the aerial parts of the perennial *Lolium* plant. Zeolite (15 g/kg) and apatites (4 g/kg soil) significantly reduce the absorption of Cd and Pb in *Zea mays* and *Hordeum vulgare* crops [453].

The benefits brought by using zeolite (50 g$^-$ kg$^{-1}$ clinoptilolite) were highlighted by Moeen, M. et al. [454] in a pot experiment that lead in the reduction of extractable heavy metals from the soil as follows: Cd, 5.51% < Pb, 23.15% < Zn, 28.41 < Cu, 35.66%. In addition, the pH and cation exchange capacity of the soil increased. The zeolite modified

with ammonium ions and calcium demonstrated good capacities to reduce Cu and Pb from the organs of *Lycopersicum esculentum* (tomato), but without efficiency on Cd and Zn, regardless of the concentration used (5–10%) [455].

Good results in reducing the amount of Cd in the soil cultivated with *Triticum aestivum* were obtained in the research carried out by Rizwan, M. et al. [456] using manure, silicon and biochar as amendments.

Biochar has an alkaline character; however, the pH largely depends on the raw material from which it is obtained and on the temperature used in the pyrolysis process. The higher the temperature, the more the pH value increases, which raises the soil pH and improves the microbial community, and through the compounds present in biochar (2-phenoxyethanol, benzoic acid, hydroxy-propionic and butyric acids, ethylene glycol and quinones) some of the pathogens harmful to plants are eliminated [457]. Phosphates and sulfates present in the ash content stabilize heavy metals by precipitation [458]. Biochar used as an amendment increases the content of C, N, K, Ca and Mg, and sequesters carbon for climate change mitigation [459]. It immobilizes Cd, Zn, and Pb, but mobilizes As and Cu in the research conducted by Lwin, C.S. et al. [429]. Sigua, G.C et al. [460] demonstrated that the use of biochar from the poultry layer and beef cattle layer obtained at temperatures of 500 °C and above had the greatest capacity to reduce the bioavailability of Zn and Cd, simultaneously with their reduction from the soil by phytostabilization (BCF > 1), and increased total biomass at the *Zea mays* crop. The effects of biochar were also emphasized in the research of Montoya, D. et al. [461], where its administration led to the enrichment in $NO_3^-$ $PO_4^{3-}$ and $SO_4^{2-}$ of bioactive compounds in *Brassica oleracea* var. *italica* flowering heads, compared to an addition of organic manure.

Depending on the raw material from which the biochar is obtained, the mobility and availability of Cu, Cd and Pb can be reduced (peat moss biochar). The availability of Cd and Zn decreases by 66% by using oak wood biochar and by 68–92% Ni and 76–93% Mn by amending with bioenergy waste biochar. In addition, it improves aggregate stability, water-holding capacity, pore-size distribution in clay soil and bulk density, total porosity, and plant-available water in sandy soils [462].

In experimental conditions carried out by Hanč, A. et al. [463], bentonite and limestone introduced as amendments in sewage sludge could reduce available Cd. Bentonite is a good sequestrant of Ni when added in doses of 5% [464]. Additionally, Na-bentonite in combination with *Festuca arundinacea* used in the decontamination of soil taken from the surroundings of a former mine and a zinc-lead smelter, which contained Pb, Cd and Zn, showed good abilities of the accumulation of large amounts of the three metals at the roots of the plant [465].

The soil with loamy texture, pH-7.8, co-contaminated with As, Pb, and Cd enriched with Fe-bentonite and Fe-zeolite, led to the conclusion that Fe-zeolite has a positive effect in the reduction of Cd and Pb in the roots and shoots of *Helianthus annus*; however, it increases the concentration of As while the application of Fe- bentonites reduces As, Cd and Pb in roots and shoots; therefore, it can represent an effective approach in soil decontamination of the three heavy metals [466].

Slag, an alkaline by-product obtained from the metallurgical industry, favors the sorption capacity of Cu in the root and decreases the foliar concentration of Cu in dwarf beans [467]. Similar results were obtained by Bes, C.M. et al. [468] with the same dwarf bean plant grown on soil contaminated with Cu by using slag, phosphates, alumino-silicates, iron grit and sewage sludge compost 5%.

Limestone, steel slag and acid mine drainage sludge administered in percentages of 3, 5 and 10% were used in the research conducted by Hong, Y.K. et al. [469]. The results highlighted the fact that all three amendments can be useful in the decontamination of soil contaminated with Pb and Cd cultivated with *Lactuca sativa*. The authors specify, however, that the most effective amendment, that considerably reduced Pb and Cd, was acid mine drainage sludge.

The decontamination of a technosol mining area rich in As and Pb was carried out by Nandillon, R. et al. [470] by adding three amendments: compost, biochar, or iron grit, alone or combined, using *Trifolium repens* as a plant. The combination of the three amendments resulted in a significant decrease in Pb concentrations in the above-ground organs, and for As the most effective treatment consisted of supplementing with 5% biochar and 5% compost.

Using a beringite rock had a strong positive effect when treating soil contaminated with Zn and Cd in Belgium [471]. The decontamination of some industrial residues contaminated with Zn and Cd, by cultivating *Brachiaria decumbens* (signal grass) and using the amendments calcium silicate (2–3%) and brewery sludge 20%, led to the reduction of the availability of Cd and Zn from the residue [472].

Although soils with a sandy texture have a more acidic character and are more suitable for Cr IV phytoextraction, the phytostabilization mechanism can be applied by adding amendments of chalcedonite, dolomite and limestone, with the results being more than promising in the combination of chalcedonite/*Festuca rubra* [435]. *Chrysopogon zizanioides* is a plant that can be used in the phytostabilization of mining areas (pH 6.8 and texture: loam) contaminated with Cr, Co, Cu and Zn, but also in the phytoextraction of Ni by administering attapulgite 2.5% [473].

Apart from these amendments, let us not forget the local amendments, which can replace the classic ones. One of these is a crushed mussel shell. The use of this amendment on the contaminated area of some mining areas and areas cultivated with vines rich in this metal, starting from a very acidic pH of the acid soil (3), demonstrated high mobilization and stabilization rates of Cu [474].

Precautions and Limitations in Phytostabilization

Despite all the advantages, phytostabilization is a less-developed mechanism of phytoremediation [471]. It lends itself to soils with, at most, moderate contamination, having several factors as the starting point of contamination such as acid or alkaline conditions, organic matter, oxygen level, and contaminant concentration (<1500 ppm Cu + Pb + Zn + As + Cd + Hg) [475,476]. This solidification/stabilization mechanism does not apply to forms of metals such as oxyanions (e.g., $Cr_2O_7$, $^{2-}$, $AsO_3$), or those that do not have low-solubility hydroxides in the category in which Hg falls [477].

In the phytostabilization mechanism, the use of organic matter is an advantage because it positively influences cationic exchange, buffering and the capacity to retain heavy metals [478]. However, over time, it can represent a limiting factor because, through decomposition, it forms various organic acids that can alter the availability of heavy metals [479] and lead to salinization, alkalinization of soil and impurity of groundwater [337]. One way to prevent surface and groundwater pollution is to ensure the phosphorus and nitrogen requirements specific to crop plants by using compost below the optimal amounts, which will only satisfy the needs of the plants [480]. Regardless of the amendments used, sewage sludge or chicken manure, leaves of *B. oleracea* var. capitata, and heads of *Brassica oleracea* var. italica have been shown to be poor accumulators of Cr, Ni, Cu, Cd and Pb; instead, the bioaccumulation factor of Zn and Mo in *B. oleracea* var. capitata is rich in soil supplementation with chicken manure, indicating a low probability that cabbage is grown on soils containing both heavy metals [481]. The results of the study conducted by Shabir, H.W. et al. [482] demonstrate that vermicompost must be used with caution, because at higher amounts (40–50 ppm) it shows adverse effects on plant growth.

The excessive use of fertilizers based on nitrogen and phosphorus affects the pH of the soil by acidifying it, which has the consequence of accelerating the absorption of Cd in plants [483]. Nitrogen amendments influence Cd assimilation capacity differently in the research conducted by Ur Rehman, M.Z. et al. [484], where *Solanum nigrum* had a higher affinity for ammonium ($NH_4$) than nitrate ($NO_3$). It has the same affinity for urea, which increases the concentration of cadmium (Cd) in shoots and roots.

Sometimes, the application of lime in doses of 5–10% negatively affects plant growth consecutively with the availability of heavy metals. Soil contaminated with Pb and Cd can induce a toxic character in cultivated plants through the accumulation of cadmium, which, together with carbonates, is released, dissolves in the rhizosphere, and is taken up by them. Although alkaline amendments lead to an increase in pH and bind heavy metals by precipitation, it is important to know the type of amendment because the application of ferrous sulfates acidifies the soil by oxidizing iron sulfate. In this situation, the simultaneous application of lime or biochar is important to prevent the remobilization of Cd, Hg, Cu and Zn [459]. The efficiency of biochar, an amendment used in phytostabilization, depends a lot on the origin of the raw material (agriculture, forestry, household, livestock waste), the physicochemical properties, the mechanism used in phytoremediation and the interaction between the microbial population, plant roots and soil particles [284]. Biochar is not applied in wet and flooding soil because the absorption of heavy metals is limited. It lends itself better to dry and arid conditions; moreover, in areas with heavy pollution, its use leads to the increased absorption of heavy metals [462]. The use of coal fly ash can have negative consequences on the microbial fauna and enzyme activity in the soil as a result of increasing the sorption of heavy metals in plants [485].

The application of lamination slag and red mud to immobilize As does not have the expected effect, and the combination of organic fertilizers and iron oxides are not recommended because it can induce plant toxicity by increasing the concentration of As in the aerial parts of plants, due to the interaction between phosphate and arsenate at soil absorption sites [464]. *Medicago sativa* lends itself to the phytostabilization of soil with a sandy clay loam texture, pH 7.6, and a high content of Pb, Zn and Cd by adding sheep manure as an amendment; instead, adding KCl, an inorganic fertilizer, increased the amount of metals in the aboveground tissues of the plant in the experimental research carried out by [486].

## 5. Conclusions and Future Perspectives

The major sources of soil pollution with heavy metals are largely a reflection of anthropogenic activities. The unprecedented intensification of industrial activities in recent decades, the use and application of treatments to combat diseases and pests in agriculture, and the fertilization of the soil with manure in order to increase productivity and reduce food shortages in certain areas of the world are real sources of air pollution.

Heavy metals accumulate at a faster rate than the depollution capacity of the soil, but these sentinels, the plants growing in the polluted areas, are valuable indicators that can represent the first step towards choosing the best phytoremediation mechanism.

In the success of phytoremediation, the conditions for choosing plants are multiple but the bioconcentration factor and the biomass obtained per hectare are essential.

In phytoextraction/phytostabilization, it is good to use plants with a wide-spread resistance to unfavorable climatic conditions and double purposes, including soil decontamination and the production of biogas, bioethanol, incineration and the possibility of extracting heavy metals. If the purpose of incineration is that of valorization by obtaining heat, plants that have the ability to produce low $CO_2$ emissions are preferable (a very topical problem).

As plants cannot decontaminate deep layers of soil, but only those at their roots, *Medicago sativa Linum usitatissimum Helianthus annuus, Lolium perenne* are part of deep-rooted plants.

Phytoextraction is suitable for soils with a lower pH and phytostabilization for those alkaline or with high salinity. Plants in phytoextraction accumulate large amounts of heavy metals in plant organelles above ground, and those used in phytostabilization sequester heavy metals at the root level and are suitable for soils which are at most moderately contaminated with heavy metals, while phytoextraction can be a useful tool on soils heavily polluted.

Soil pH, organic matter and the amount of clay in the soil play essential roles in the phytoremediation mechanism, because many metal cations are more soluble and available in solution at a low pH. Cd and Zn ions have a relatively low affinity for organic matter, while Pb and Cu have a strong affinity for organic matter.

There is a wide range of chelating agents used in induced phytoextraction that have different affinity for heavy metals. EDTA for Pb, however, is non-biodegradable, can contaminate groundwater and, depending on the time of treatment, can affect germination, and biomass obtained EDSS has an affinity for Cu and Zn; less for Pb, but it is biodegradable and does not affect biomass. Replacing non-biodegradable agents with natural ones remains the most feasible option. Citric acid used in the phytoextraction of Cu, Zn and U has good results and the amount used is not restricted.

In phytostabilization, there are numerous organic amendments, for example mineral absorbents, zeolites, chalcedonite besides limestone, and dolomites, apatites, manure, silicon, biochar, bentonites, Fe-zeolites, slag, iron grit, beringite rock and crushed mussel shell, with sequestration effects on heavy metals in plant roots. Most plants are grasses, but even plants recognized as hyperaccumulators can participate in this mechanism.

Future research should focus on local or widespread plants that are not considered invasive and that correspond to the specific pedo-climatic conditions of polluted areas, and that are able to produce certain organic compounds under stress conditions, and which have the ability to bind heavy metals without polluting the water and the environment, and without changing the biotope of the respective areas.

**Author Contributions:** Conceptualization, C.H.; methodology, C.H., S.-N.P. and A.-S.R.; software L.A. and A.-S.R.; data curation C.H., A.-A.C., D.S.D. and A.-S.R.; resources, C.H., L.A., S.-N.P., A.-A.C., D.S.D. and A.-S.R.; writing—original draft preparation C.H. and S.-N.P.; writing—review and editing C.H. and S.-N.P.; visualization, C.H., L.A., S.-N.P., A.-A.C. and D.S.D. All authors have read and agreed to the published version of the manuscript.

**Funding:** This research received no external funding.

**Data Availability Statement:** The data presented in this study are openly available in research-gate.net/. The data presented in this study are available. The data were taken and processed from articles with free access.

**Conflicts of Interest:** The authors declare no conflict of interest.

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
