# Peer review of "The Journey of 1000 Leagues towards the Decontamination of the Soil from Heavy Metals and the Impact on the Soil–Plant–Animal–Human Chain Begins with the First Step: Phytostabilization/Phytoextraction"

_agriculture, doi:10.3390/agriculture13030735_

Round 1
Reviewer 1 Report
The study is complex and well documented. I recommend this paper be accepted and published in this journal. However, there are some recommendations regarding this manuscript.
1. As a review paper, the paper quoted many documents, but the paper is too long to be like a book. Some common-sense contents only need to be briefly described, and it is recommended not to elaborate. In general, it is suggested to shorten the article.
2. There are many element symbols. It is recommended to use words for the first time, and then use element symbols directly,eg. Cadmium (Cd) to Cd.
3.Conclusion - This part it is just a repeating of the earlier information, only the most important observations and conclusions from the research should be included here.
Author Response
First of all, I want to thank you for the time allocated to the revision of this manuscript, which was very carefully read by each of the reviewers. I appreciate! As much as possible I have tried to touch on every sub point of the parts that you felt were not correct. In order to be able to fit in and answer the sub-points and all the reviewers, the paper is slightly restructured, in the sense that the introduction part has been shortened and only one table has remained compared to the original manuscript.
In order to reach a compromise that would satisfy all the requirements of all the reviewers, I reduced the text of the first part, we made the changes related to the elimination of the word and left only the symbols, the part of the conclusions has been redone.
Thank you,
Best regars, Cristina Hegedus

Reviewer 2 Report
Comments to authors:
I have revised your manuscript entitled “The journey of 1000 leagues towards the decontamination of the soil from heavy metals and the impact on the chain: soil-plant-animal-human. begins with the first step: phyto-stabilization/phytoextraction” This is an interesting topic. But it needs many important changes.
Note: Authors should insert the line and page numbers when they carry the corrections.
· What is the novelty of this review article, although there is a lot of literature, manuscripts, and books on this point?
· Definition the “soil pollution” and “soil contamination” as well as what is the difference between them.
· Insert Factors affecting the availability of these elements.
· Page 2 Line 50: “(> 5)” correct to “(> 5 g cm-1)”
· Page 2 Line 60: remove Cobalt (Co), Because it is not considered from micronutrients
· Page 2 Line 66: remove “or human activities”
· Page 2 Line 69-73: “The type of vegetation,………………………… contribute to soil pollution” What does this have to do with Natural sources of soil pollution? The authors should focus on the Natural sources of soil pollution
· Line 107: “mints” what the meaning of.
· In Table 1: “(g/ha)” What is this unit used for.
· Line 137: replace “extracted” by “Manufactured"
· Line 153: “stibium (Sb)” correct to “Antimony(Sb)”
· Line 217: “reactive proteins (ROS)” It is the same abbreviation “reactive oxygen species (ROS)” existence in Line 191.
· Line 222: “concentration” For whom is this.
· Line 266: “metals heavy” correct to “heavy metals”
· The quality of the tables is not good. It should be well modified to make it easier to read.
· Line 500: “Salt, D.E. and Salt, D.E.” is an error.
· The authors should make a comparison between decontamination methods in terms of benefits, harms and costs, as this paper has scientific value.
· Line 759: “cooper” correct to “copper”
· Line 791: “DDT” what the meaning of
· The authors should definition of the terms include in the review article such as phytostabilization, phytoremediation, …………………….
· Line 818: is oxides and hydroxides of lead (Pb) soluble in water.
· Line 956: “Kg” correct to “kg”
· Line 1083: “GLDA (N,N-dicarboxymethyl glutamic acid)” what the meaning of this abbreviation GLDA.
· The units in this review article “mM , mmol/kg, mg/kg, mg kg–1 , 1 mg Hg/g , and mg g-1“ they not unity. Authors must standardize units.
· Line 1205: “NLMWOA” correct to “Natural low molecular weight organic acids (NLMWOA)”
· Authors must, for plants, write the English name and the scientific name in brackets like this ryegrass (Lolium perenne L) in all review article.
· Line 1483: “g-“ is an error
· Line 1511: “NO-3” is an error
· Line 1579: correct that “e.g., Cr2O7, 2-, AsO3)”
· Please rewrite the conclusions section in an understandable and smooth manner.
Author Response
First of all, I want to thank you for the time allocated to the revision of this manuscript, which was very carefully read by each of the reviewers. I appreciate! As much as possible I have tried to touch on every sub point of the parts that you felt were not correct. In order to be able to fit in and answer the sub-points and all the reviewers, the paper is slightly restructured, in the sense that the introduction part has been shortened and only one table has remained compared to the original manuscript
In order to reach a compromise that would satisfy all the requirements of all the reviewers, I reduced the text of the first part, made the changes related to the elimination of the word and left only the symbols, the part of the conclusions has been redone, the names in Latin were introduced and were used until the end of the manuscript. The bibliography has been fully translated.
It is true that there are a multitude of books, chapters, papers, related to this current topic, but from our point of view we tried to approach as analytically as possible both sides of the benefits/limitations coin; to insist on plants with a wide spread, resistant to climatic stress or that induced by the presence of heavy metals and whose benefits extend their area and can be used for another purpose
I cannot tell you exactly the line where corrections were made because each page starts again with 1, but I can tell you the page and and the changes are red in manuscript
line 50 pg 2
line 60 pg 2. Cobalt for plants is removed from the list of essential micronutrients
Cobalt for plants is removed from the list of essential micronutrients
Page 2 line 69-73. I completely agree with you I removed type of vegetation from 1.1. Natural sources of soil pollution
Line 107. Mints= mint (coin) (pg 2-3) Thank you
Table 1 =Table 1,3,4 was removed (pg 5-6)
Line 137 extracted was replaced with Manufactured (pg 3)
Line 153 stibium was replaced with Antimony (Sb) (pg 3)
Line 217-222 .Since one of the reviewers thought that part 1 was too long, this part was modified, the changes were made following the suggestion of the reviewers who were much more vigilant (pg 4 and pg 20).
Line 500: removed
Line 1083 . GLDA - N,N-dicarboxymethyl glutamic acid , after references 379
Line 1205 NLMWOA was revised (pg 19, 21)
One of the reviewers suggested me to use Latin plant names until the end of the manuscript
Like a critical point, which I could not touch, was the one related to the units of measure.Thank you
Best regard, Cristina Hegedus

Reviewer 3 Report
The volume of processed material is enormous. Congratulations!
To emphasize the phytoremediation approach, I propose reducing the first part of the work by excluding repetitions or chapters 4.4 and 4.5.
The article represents a vast analytical work. Therefore, to ensure a logical and coherent presentation, it is necessary to structure the analyzed data based on the analyzed element, its harmful effect, or its mechanism of action.
When presenting a series of elements, only one form of writing is required: either the chemical name of the element or its symbol. In analyzing the effect of the element or its dispersion in soil, etc., its name and symbol may be inserted without insisting on both forms.
The text should be written following the requirements for nominalizing species (line 264, 311, 331, 612, 671, 787, etc.).
Plant species will be written in Latin in scientific publications, and their common name may be mentioned once or twice without repeating it each time the plant species is mentioned.
In chapter 4.1, "Phytoremediation and soil decontamination mechanisms," if the topic is phytoremediation, data regarding mechanisms specific to bacteria, including Cyanobacteria and fungi, which are not part of the plant kingdom, will be excluded from the text. Alternatively, the chapter title may be modified.
The reason why hydrocarbon was nominated in the list of pollutant elements - line 453.
What does "reduce energy levels" mean - line 347.
The term "plant membranes" is incorrect.
In Table 2, the common name column of the plant should be excluded. The Latin name of the plant species in column 1 may be supplemented with the common name.
In Tables 2 and 3, the animal names should be unified: either the Latin name of the species or the common name should be used.
References in Romanian will be rewritten in English with the note (in Romanian). Ref. 18, 37.
Author Response
First of all, I want to thank you for the time allocated to the revision of this manuscript, which was very carefully read by each of the reviewers. I appreciate! As much as possible I have tried to touch on every sub point of the parts that you felt were not correct. In order to be able to fit in and answer the sub-points and all the reviewers, the paper is slightly restructured, in the sense that the introduction part has been shortened and only one table has remained compared to the original manuscript.
In order to reach a compromise that would satisfy all the requirements of all the reviewers, I reduced the text of the first part, made the changes related to the elimination of the word and left only the symbols, the part of the conclusions has been redone, the names in Latin were introduced and were used until the end of the manuscript. The bibliography has been fully translated.
I cannot tell you exactly the line where corrections were made because each page starts again with 1, but I can tell you:
line 453 The reason why hydrocarbon........ has been removed (pg 12 - Limitations of phytoremediation) Line 347 Reduce energy levels ...... has been removed
subchapter 4.1 (now 3.1) where I was asked to remove the first part, which was only a brief introduction (pg 8) was partially removed.
Plant membranes replaced with cell membranes (Pg 9).
Thank you for the information about the table. Your instructions helped me a lot
Thank you
Best regard, Cristina Hegedus

Round 2
Reviewer 3 Report
I recommend the work for publication
Good luck!!